# Regulation of retinal axon growth by secreted Vax1 homeodomain protein

**Namsuk Kim[1], Kwang Wook Min[1], Kyung Hwa Kang[2], Eun Jung Lee[1], Hyoung-Tai Kim[1], Kyunghwan Moon[1], Jiheon Choi[1], Dai Le[1], Sang-Hee Lee[1], Jin Woo Kim[1,2]***

[1]Department of Biological Sciences, Korea Advanced Institute of Science and Technology, Daejeon, Republic of Korea; [2]KAIST Institute of BioCentury, Korea Advanced Institute of Science and Technology, Daejeon, Republic of Korea

**Abstract** Retinal ganglion cell (RGC) axons of binocular animals cross the midline at the optic chiasm (OC) to grow toward their synaptic targets in the contralateral brain. Ventral anterior homeobox 1 (Vax1) plays an essential role in the development of the OC by regulating RGC axon growth in a non-cell autonomous manner. In this study, we identify an unexpected function of Vax1 that is secreted from ventral hypothalamic cells and diffuses to RGC axons, where it promotes axonal growth independent of its transcription factor activity. We demonstrate that Vax1 binds to extracellular sugar groups of the heparan sulfate proteoglycans (HSPGs) located in RGC axons. Both Vax1 binding to HSPGs and subsequent penetration into the axoplasm, where Vax1 activates local protein synthesis, are required for RGC axonal growth. Together, our findings demonstrate that Vax1 possesses a novel RGC axon growth factor activity that is critical for the development of the mammalian binocular visual system.

## Introduction

Development of the mammalian binocular visual system requires topographic synaptic connections of retinal ganglion cell (RGC) axons to neurons on the lateral geniculate nucleus and superior colliculus of the brain (*Lemke and Reber, 2005*). To access their synaptic targets, RGC axons exit from the retina and grow in selective directions by recognizing guidance cues expressed in optic pathway structures, including the optic disc (OD), optic stalk (OS), optic chiasm (OC), and optic tract (OT) (*Petros et al., 2008*). RGC axon guidance cues include cell surface ligands such as semaphorins in the OS and ephrinB2 in the OC, and soluble factors such as netrin-1 in the OD and Slit1 in areas surrounding the OC (*Erskine and Herrera, 2007*). Ultimately, only about 3% of mouse RGC axons, which originate from the ventral and temporal part of the retina, are linked to targets on the same side of the brain, whereas a majority of RGC axons are connected to those on the opposite side after crossing the midline at the OC, located at the ventral-medial hypothalamic (vHT) area.

Subsets of vHT cells therefore express molecules that determine the directionalities of RGC axons at the OC. It has been shown that vHT radial glial cells express ephrinB2, which binds to EphB1 receptors expressed in ventral-temporal RGC axons and repels the axons toward the ipsilateral optic tract (*Nakagawa et al., 2000*; *Williams et al., 2003*). In addition, vascular endothelial growth factor 164 (VEGF164), an isoform of the vascular endothelial growth factor VEGF-A (*Soker et al., 1996*), and neuronal cell adhesion molecule (NrCAM) expressed in the vHT have been suggested to support the growth of RGC axons across the vHT midline by binding to neuropilin-1 and plexin-A1, respectively (*Williams et al., 2006*; *Erskine et al., 2011*; *Kuwajima et al., 2012*). To receive the directional guidance of these molecules at the vHT, RGC axons must pass through the ventral-lateral diencephalic area, where repulsive guidance cues, such as Slit and semaphorins, are expressed at high levels (*Erskine et al., 2000*; *Plump et al., 2002*). However, the molecules that support RGC axon growth toward the vHT midline are still unknown.

**\*For correspondence:**
jinwookim@kaist.ac.kr

**Competing interests:** The authors declare that no competing interests exist.

**eLife digest** We see the world around us when light bounces off of objects and hits the retina at the back of our eyes. This triggers electrical signals in neurons called retinal ganglion cells (RGCs), which have long structures called axons that extend out from the retina and into the parts of the brain where the signals are interpreted. As the axons grow, various 'guidance' molecules direct the axons to the correct part of the brain.

One molecule that is important for the growth of retinal ganglion cells' axons is a protein called Vax1. This protein is a transcription factor and binds to DNA to control how and when the molecular templates used to make proteins are made—a process called transcription. Vax1 is not produced in retinal ganglion cells, but it does control the extension of these cells' axons into part of the brain called the ventral hypothalamus. In this study, the axons cross to the other side of the brain by forming a structure called optic chiasm. Humans and mice lacking Vax1 are unable to develop the optic chiasm, and the axons of their retinal ganglion cells do not reach their targets in the brain. These defects were thought to occur because the guidance molecules whose transcription is normally controlled by Vax1 were not produced in the correct amounts when Vax1 is absent.

Kim et al. now challenge this view by creating a mutant version of Vax1 that cannot bind to DNA or regulate the transcription of other proteins. Retinal ganglion cell axons could still grow correctly when they were put close to cells expressing this version of the Vax1 protein. This contradicts a hypothesis that Vax1 supports axonal growth by transcribing guidance molecules. Kim et al. followed up these results by examining developing mice and reached the unexpected conclusion that Vax1 is secreted from cells in the ventral hypothalamus and binds to a type of sugar molecule found on the surface of the axons. Once bound, Vax1 can enter the axons where it appears to stimulate the production of proteins inside axons, which helps the axons to grow.

These findings reveal unconventional functions for Vax1 that occur in addition to its role as a transcription factor. Vax1 is known to regulate the development of several structures in the brain, so the work of Kim et al. also raises new questions about how Vax1 controls these processes.

Ventral anterior homeobox 1 (Vax1) is a homeodomain transcription factor expressed in various ventral-medial forebrain-derived structures, including the medial and lateral geniculate eminences, the ventral septum, the anterior entopeduncular area, the preoptic area, the vHT, and the OS (*Hallonet et al., 1998*; *Bertuzzi et al., 1999*). Genetic inactivation of *Vax1* in humans and mice causes agenesis of multiple midline structures of the brain, including the anterior commissure, the corpus callosum, and the OC, in addition to the coloboma of the eye (*Bertuzzi et al., 1999*; *Hallonet et al., 1999*; *Slavotinek et al., 2012*). RGC axons in *Vax1*-deficient mice can grow through the OS but cannot access the vHT area and fail to form an OC. *Vax1* is not expressed in RGCs despite its critical roles in growth and fasciculation of RGC axons (*Bertuzzi et al., 1999*; *Mui et al., 2005*). Therefore, it has been suggested that defects in OC formation in *Vax1*-deficient mouse RGCs might be caused by the loss or gain of axon guidance cues that are potential transcription targets of Vax1 in vHT cells.

Contrary to expectation, we here found that Vax1 promoted RGC axon growth in a transcription-independent manner. Moreover, Vax1 is secreted from vHT cells and binds and enters RGC axons to stimulate axonal growth. This unexpected trafficking of Vax1 to RGC axons was mediated by heparan sulfate proteoglycans (HSPGs), such as syndecan and glypican, expressed in RGC axons. However, Vax1 binding to HSPGs was not sufficient to trigger RGC axon growth; penetration into the RGC axoplasm and subsequent stimulation of local protein synthesis were also necessary. Taken together, our findings reveal an unconventional function of Vax1 as an RGC axon growth factor that enables RGC axons to grow toward the midline during development.

## Results

### Vax1 regulates the RGC axonal growth in a non-cell autonomous manner

*Vax1* is expressed in cells located in optic pathway structures, such as the OS and vHT, and plays an essential role in fasciculation of RGC axons and formation of the OC (*Bertuzzi et al., 1999*; *Hallonet et al., 1999*). At the vHT of days post coitum 14.5 (E14.5) mouse embryo, Vax1 is expressed in

Sox2 (SRY box 2)-positive neural progenitor cells (NPCs) and RC2-detectable nestin-positive radial glia (*Figure 1A,B*; top rows), which is known to provide RGC axon guidance cues (*Petros et al., 2008*). Although the morphology of the chiasm is abnormal in *Vax1*-deficient (*Vax1⁻/⁻*) mice (*Bertuzzi et al., 1999*), these OC-forming cells and several chiasm-localized cues (NrCAM and *Vegfa*) are still present (*Figure 1A,B*, bottom rows; *Figure 1—figure supplement 1*). However, vHT explants from *Vax1⁻/⁻* mice were unable to attract RGC axons regardless of the *Vax1* gene status of co-cultured retinal explants, whereas wild-type (WT; *Vax1⁺/⁺*) vHT explants were able to attract RGC axons projected from *Vax1⁻/⁻* explants as well as WT explants (*Figure 1C,D*). We therefore concluded that Vax1 controls

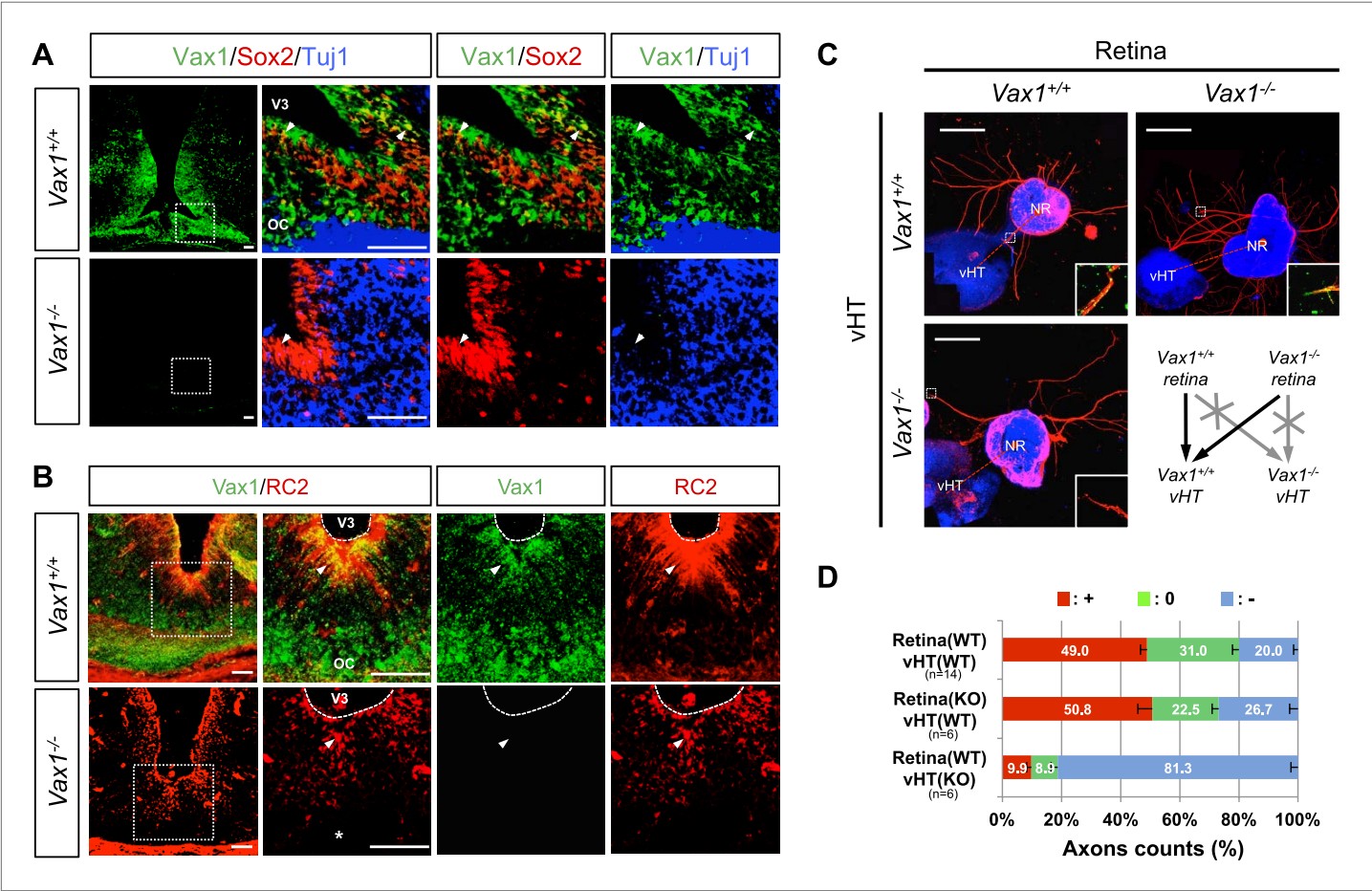

**Figure 1**. Vax1 regulates RGC axonal growth in a non-cell autonomous manner. (**A**) Cells expressing Vax1 (green) in brain sections (coronal; 16 μm) from E14.5 *Vax1⁺/⁺* (top) and *Vax1⁻/⁻* (bottom) embryos were detected by co-immunostaining for the NPC marker Sox2 (red) and post-mitotic neuronal marker tubulin-βIII (blue), detected with the Tuj1 antibody. The right-most three columns are the magnified images of dotted boxes in the left column image. The results indicate that Vax1 is expressed in a subpopulation of Sox2-positive NPCs (arrowheads) but is not detectable in Tuj1-positive neurons. (**B**) Vax1-expressing cells in the vHT were also compared with RC2-positive radial glia. Arrowheads indicate RC2-positive radial glial cells expressing Vax1. Scale bars: 50 μm. (**C**) The vHT and dorsal neural retina (NR) were isolated from WT (*Vax1⁺/⁺*) and *Vax1*-knockout (*Vax1⁻/⁻*) E13.5 mouse embryos and co-cultured in a combinatorial manner for 48 hr. The explants were fixed and immunostained with an anti-NF160 antibody (α-NF160; red); nuclei were counterstained with DAPI (blue). Dotted boxes indicate the area magnified in each inset. Red dotted lines link centers of retinal explants and vHT explants. Scale bars: 500 μm. (**D**) The angular distribution of RGC axons in images was measured by counting pixels containing immunostaining for the axon marker NF160 (axon counts), as described in 'Materials and methods', and presented graphically. +, forward direction angle segment; 0, neutral direction angle segments; −, reverse direction angle segment. The values in the bar are averages, error bars denote standard deviations (SDs), and numbers under *y*-axis labels are the numbers (n) of explants analyzed from three independent experiments. *p*-values determined by the analysis of variance (ANOVA) are between 0.01 and 0.005.

The following figure supplement is available for figure 1:

**Figure supplement 1**. Expression of RGC axon attractive cues in *Vax1⁻/⁻* mouse vHT.

the RGC axonal growth in a non-cell autonomous manner, potentially by regulating the expression of unidentified secreted axon-guidance molecules.

## Vax1 is a secreted protein

To identify Vax1-regulated secreted factors that control RGC axonal growth from co-cultured retinal explants, we overexpressed mouse Vax1 in COS7 cells. RGC axons from retinal explants grew preferentially toward co-cultured Vax1-expressing COS7 cell aggregates, whereas RGC axons projected in random directions upon co-incubation with untransfected or Vax2-overexpressing COS7 cell aggregates (*Figure 2A,B*). Because Vax2 shares an identical homeodomain with Vax1 (*Barbieri et al., 1999*), these results indicate that the RGC axon growth stimulatory activity is specific for Vax1.

We next examined whether Vax1-induced RGC axonal growth is dependent on Vax1 transcription activity by co-incubating retinal explants with COS7 cells expressing a transcriptionally inactive Vax1(R152S) mutant (*Figure 2*, *Figure 2—figure supplement 1*). This mutation was reported in a human patient who exhibited coloboma, cleft palate, and agenesis of corpus callosum (ACC), phenotypic manifestations similar to those of *Vax1$^{-/-}$* mice (*Slavotinek et al., 2012*). Unexpectedly, we found that Vax1(R152S)-expressing COS7 cells were also able to induce RGC axonal growth as efficiently as WT Vax1-expressing COS7 cells (*Figure 2A*, third row, B), suggesting that Vax1 induces RGC axonal growth in a transcription-independent manner.

More strikingly, Vax1 and Vax1(R152S) proteins were not only expressed in transfected COS7 cells, they were also detectable in neurofilament 160 kDa (NF160)-positive RGC axons projecting from co-cultured retinal explants (*Figure 2A*, right two columns). These axonal Vax1-immunostaining signals were remarkably decreased in the presence of a rabbit anti-Vax1 polyclonal antibody (α-Vax1) that sequesters Vax1 in the growth medium (*Figure 2—figure supplement 2*). Furthermore, Vax1 and Vax1(R152S) proteins were found in the growth medium of transfected COS7 cells, whereas Vax2 protein was not (*Figure 2C*). Since the viability of the transfected COS7 cells were not different from each other (data not shown), these results suggest that Vax1 proteins in the growth medium and co-cultured RGC axons did not originate from dead cells. Similar to overexpressed Vax1 in COS7 cells, endogenous Vax1 expressed in vHT explants was detectable in the growth medium (*Figure 2D*). Furthermore, Vax1 protein was also identified in the cerebral spinal fluid (CSF) of E14.5 mouse embryos (*Figure 2E*), suggesting that Vax1 is secreted in vivo as well as in vitro.

## Vax1 has retinal axon growth factor activity

We further tested whether secreted Vax1 is capable of directly binding to RGC axons and regulating axonal growth using purified recombinant Vax1 protein. Time-lapse recordings of RGC axons revealed that fluorescein isothiocyanate (FITC)-labeled, His-tagged Vax1 (Vax1-His) protein added to the growth medium of retinal explants accumulated in RGC axons and exerted strong growth stimulatory effects on them (*Figure 3A,B*; *Videos 1–3*; *Figure 3—figure supplement 1*). The axon growth stimulating effects of Vax1-His were applied equally to retinal quadrants (*Figure 3—figure supplement 2*), implicating Vax1 is not a region-specific axon growth factor. The transcriptionally inactive Vax1(R152S)-His mutant protein was also detectable in RGC axons and stimulated axonal growth as efficiently as WT Vax1-His (*Figure 3C,D*), suggesting that extracellular Vax1 moves to RGC axons and induces axonal growth in transcription-independent manner. Despite that Vax2 is not secreted (*Figure 2C*), the ability of recombinant Vax2-His to be internalized and induce RGC axonal growth is almost equivalent to that of Vax1 (*Figure 3—figure supplement 3*), implicating that internalization but not secretion is a conserved characteristic of VAX transcription factors.

## In vivo evidence for intercellular transfer of Vax1

We next sought evidence for the transfer of Vax1 to RGC axons in vivo. As reported previously (*Hallonet et al., 1998*; *Bertuzzi et al., 1999*), *Vax1* mRNA is expressed in RGC axon-associated structures, including the OD, the OS, and the vHT, but not the retina, of E14.5 mice (*Figure 4A*, *Figure 4—figure supplement 1A*). However, an examination of Vax1 protein distribution in E14.5 *Vax1$^{lacZ/+}$* heterozygous mouse embryos showed that Vax1 was detectable in the retina as well as the OS and OD (*Figure 4B*, top row). In these mice, β-galactosidase (β-gal) is expressed from a *lacZ* gene replacing one *Vax1* gene locus while Vax1 is expressed from the other intact *Vax1* gene locus (*Hallonet et al., 1999*); therefore, β-gal should be expressed in cells expressing Vax1. However, we found that RGCs in *Vax1$^{lacZ/+}$* mice did not express β-gal but did express Vax1, this contrasts with OS APCs which co-expressed Vax1 and β-gal (*Figure 4B*, top row). Vax1-immunostaining signals were completely absent in RGCs as well as

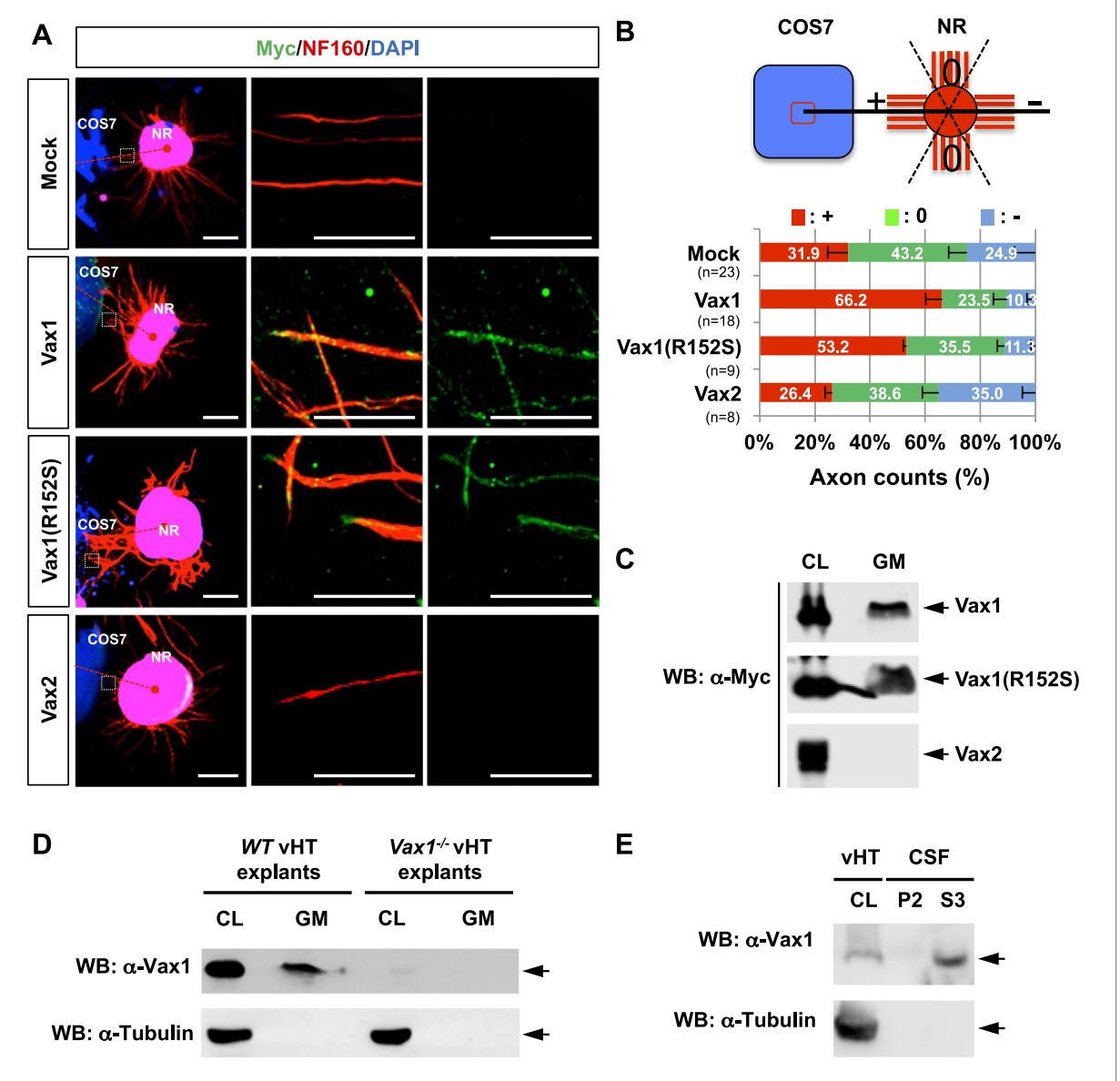

**Figure 2**. Vax1 homeodomain protein is a secreted protein. (**A**) COS7 cells overexpressing Myc-tagged mouse Vax1, Vax1(R152S), or Vax2 were co-cultured with E13.5 mouse retinal explants (NR) for 48 hr. The explants were then stained with a rabbit anti-Myc antibody (green) and a mouse anti-NF160 antibody (red). Nuclei of explant cells were counterstained with DAPI (blue). Dotted red lines indicate the connections between the centers of two explants. Scale bars: 500 μm (left column) and 100 μm (magnified immunostained images in two right-hand columns). (**B**) The angular distribution of RGC axons was measured as described in *Figure 1D*. The values in the bar are averages and error bars denote SDs. Numbers under *y*-axis labels are the number of explants analyzed from three independent experiments. *p*-values are between 0.01 and 0.005 (ANOVA). (**C**) Growth medium from COS7 cells overexpressing Myc-Vax1, Myc-Vax1(R152S), or Myc-Vax2 was collected, and the presence of Vax protein in the growth medium (GM) was detected by Western blotting (WB) with an anti-Myc antibody. The relative amounts of secreted protein were also measured by analyzing the level of proteins in the COS7 cell lysates (CL; 5% of total). (**D**) vHTs isolated from WT and *Vax1^{−/−}* E13.5 mouse embryos were cultured for 24 hr, after which GM was collected for detection of secreted Vax1 protein by Western blotting. CL, cell lysates of vHT explants (5% of total). (**E**) Cerebrospinal fluid (CSF) from E14.5 mouse embryos (n = 20) was collected, a supernatant fraction (S3) was separated from cell debris (P2) by step-wise centrifugation (see 'Materials and methods' for details), and the presence of Vax1 protein was examined by Western blotting. The presence of β-tubulin, a cytoplasmic protein, was also examined in GM and CSF fractions to check for the possible non-specific release of intracellular proteins from dead cells. CL, E14.5 vHT cell lysates (2% of total lysates from one embryonic vHT).

The following figure supplements are available for figure 2:

**Figure supplement 1**. Relative transcriptional activities of Vax1 mutants used in this study.

**Figure supplement 2**. Interference of Vax1 intercellular transfer by sequestering extracellular Vax1.

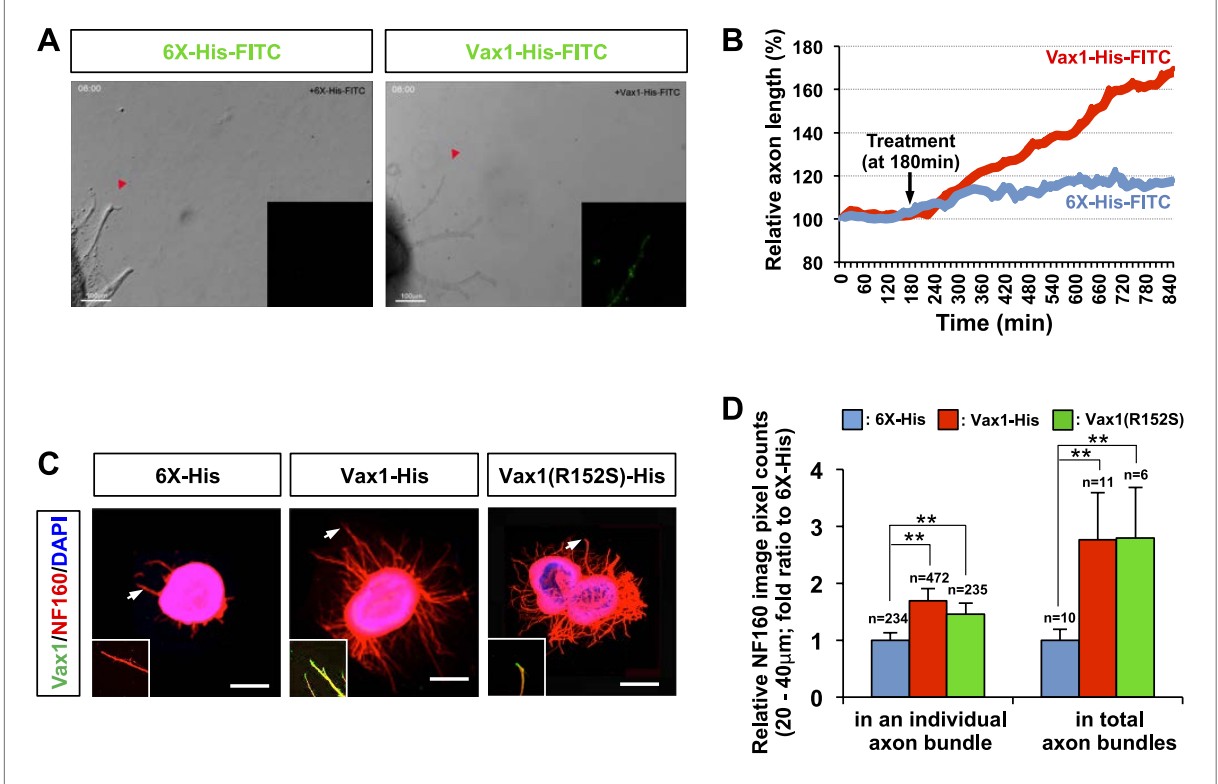

**Figure 3**. Vax1 protein is a retinal axon growth factor. (**A**) E13.5 mouse retinal explants were cultured for 24 hr and then treated with 6X-His-FITC peptide (100 ng/ml) or recombinant Vax1-His-FITC protein (500 ng/ml) for an additional 24 hr. Images of RGC axons were taken every 15 min for 16 hr before immunostaining with anti-Vax1 and anti-His antibodies (*Videos 1 and 2*; *Figure 3—figure supplement 1*). The accumulation of 6X-His-FITC and Vax1-His-FITC in growing RGC axons was also visualized by detecting FITC fluorescence signals (inset images). Red arrowheads indicate the area magnified in each inset. (**B**) The changes in RGC axonal length during the recording were plotted after adjusting the initial length to 100%. (**C**) Retinal explants treated with 6X-His (25 ng/ml), Vax1-His (100 ng/ml), or Vax1(R152S)-His (100 ng/ml) for 24 hr were stained with rabbit anti-Vax1 (green) and mouse anti-NF160 (red) antibodies to visualize Vax1 protein in RGC axons. Arrowheads indicate the area magnified in each inset. Scale bars: 500 μm. (**D**) Relative numbers of axon bundles projecting from retinal explants were indirectly measured by counting the pixels containing NF160 immunofluorescence in RGC axons between 20 and 40 μm from the rim of the explants (total axon bundle). The relative thickness of individual axon bundles was also measured by comparing the total pixel counts of NF160 in the 20–40-μm area (individual axon bundle). The values in the graph are averages expressed relative to those of 6X-His peptide-treated samples, presented as 1; error bars denote SDs (**p < 0.001; ANOVA). The scores on top of the graph columns are the number of axons (individual axon bundle) and the number of explants (total axon bundle) analyzed, respectively. Results were obtained from three independent experiments. The number of explants analyzed: for 6X-His, n = 10; Vax1-His, n = 11; Vax1(R152S)-His, n = 6. (already shown in total axon bundle).

The following figure supplements are available for figure 3:

**Figure supplement 1**. Penetration of exogenous Vax1 protein into RGC axons.

**Figure supplement 2**. Region non-selective stimulation of retinal axonal growth by recombinant Vax1.

**Figure supplement 3**. Recombinant Vax2 is capable for inducing RGC axon growth in vitro.

β-gal-positive OS APCs from homozygous *lacZ* knock-in (*Vax1^{lacZ/lacZ}*) *Vax1^{lacZ/lacZ}* littermates, suggesting that Vax1 immunostaining signals in the *Vax1^{lacZ/+}* mouse RGC were specific (*Figure 4B*, bottom row). Collectively, these data demonstrate that Vax1 protein in RGCs might originate from the neighboring Vax1/β-gal double-positive APCs in the OS or NPCs and radial glia in the vHT (*Figure 4B*, *Figure 4— figure supplement 1B*).

Vax1 protein in OS APCs was present mainly in nuclei, whereas a majority of Vax1 protein in β-gal-negative RGCs was non-nuclear (*Figure 4B*, i and ii). Furthermore, Vax1 co-localized with NF160 in E14.5 WT mouse RGC axons but was lost in *Vax1^{lacZ/lacZ}* mouse RGC axons (*Figure 4C*). These Vax1 localization patterns in the OS APCs and RGCs were further confirmed by immuno-transmission electron microscopy.

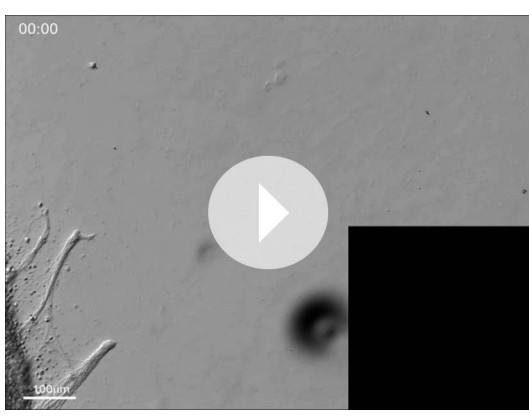

**Video 1**. Time-lapse video of E13.5 mouse retinal explants cultured in the presence of 6X-His-FITC peptides (100 ng/ml) as described in 'Materials and methods'.

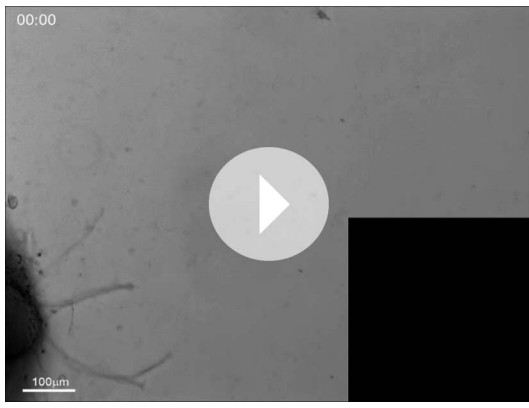

**Video 2**. Time-lapse video of E13.5 mouse retinal explants cultured in the presence of Vax1-His-FITC proteins (500 ng/ml) as described in 'Materials and methods'.

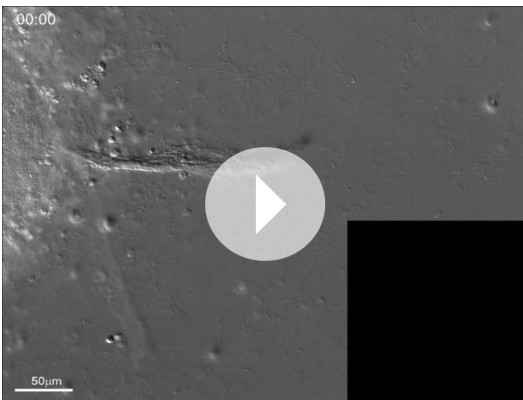

**Video 3**. Time-lapse video of E13.5 mouse retinal explants cultured in the presence of 6X-His-FITC peptides (500 ng/ml) as described in 'Materials and methods'.

Vax1 protein was highly enriched at the extracellular part of the RGC membrane and was also detected in the cytoplasm and intracellular vesicles (*Figure 4D*, top row; *Figure 4—figure supplement 2*). Vax1 protein in OS APCs was not only enriched in nuclei but was also detectable in cytoplasmic membrane structures (*Figure 4D*, bottom row). These results therefore suggest that OS- and/or vHT-secreted Vax1 might enter RGCs after docking with the RGC axon membrane.

## Intercellular transfer of Vax1 is necessary for RGC axonal growth

To investigate whether the secreted Vax1 is necessary for the growth of RGC axons toward vHT explants, we sequestered extracellular Vax1 using α-Vax1. α-Vax1 not only interfered with the transfer of Vax1 from vHT cells to RGC axons, it also antagonized RGC axonal growth toward vHT explants (*Figure 5A*, center). In contrast, neither rb-IgG (pre-immune rabbit IgG) nor α-Vax2 influenced Vax1 transfer into RGC axons or RGC axonal growth toward vHT explants (*Figure 5A*, left and right). α-Vax1 treatment not only reduced the population of retinal axons growing toward the vHT (*Figure 5B*), it also decreased the total number of retinal axons growing from the explants (*Figure 5C*), suggesting an axogenic activity as well as an axon growth-stimulating activity of extracellular Vax1.

The roles of extracellular Vax1 in RGC axon growth were also investigated in vivo. Collagen gels releasing rb-IgG or α-Vax1 were implanted in the third ventricle of E13.5 mouse embryonic brain slabs to sequester extracellular Vax1 in the vHT area (*Figure 5D*, diagram on top panel). Mouse embryos implanted with α-Vax1-releasing gels showed a remarkable reduction in RGC axons accessing the midline compared with embryos implanted with rb-IgG-releasing gels (*Figure 5D*). In α-Vax1-implanted mouse embryos, a significant number of RGC axons showed reduced Vax1 immunoreactivity and stopped at the lateral wall of the ventral diencephalon, properties similar to those observed in *Vax1*⁻/⁻ mice (*Bertuzzi et al., 1999*) (*Figure 5D*, bottom row). Taken together, these results suggest that extracellular Vax1 is necessary for RGC axonal growth to the ventral midline.

## Heparan sulfate proteoglycan regulates Vax1 intercellular transfer

Intercellular transfer has also been reported for other homeodomain transcription factors, such as engrailed-2 (En2) and orthodenticle homeobox

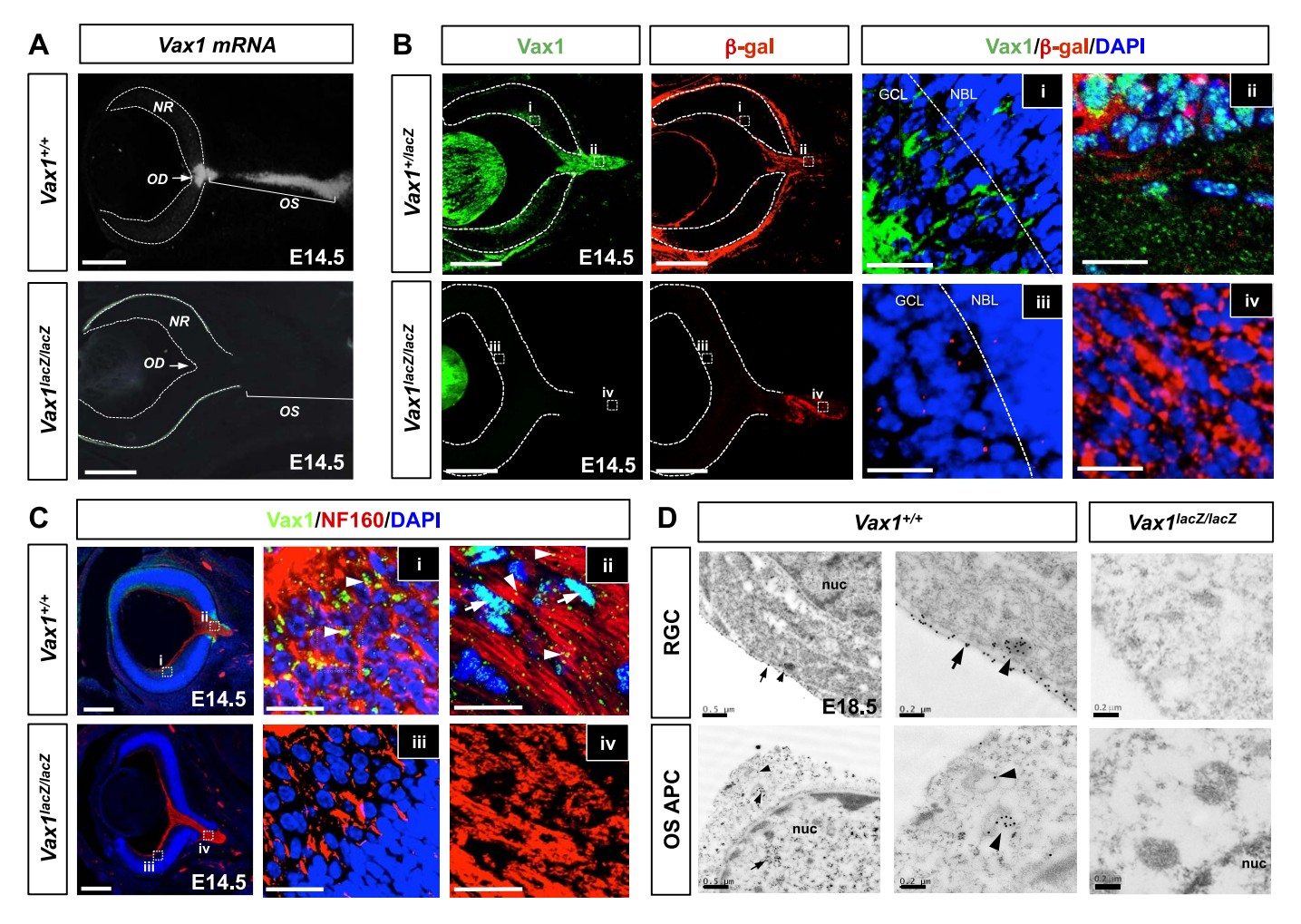

**Figure 4**. Mouse RGC axons import Vax1 protein. (**A**) *Vax1* mRNA expression in E14.5 WT (*Vax1⁺/⁺*) mouse retinas was examined by in situ RNA hybridization using a [³³P]-CTP-labeled antisense Vax1 probe, as described elsewhere (**Mui et al., 2005**). *Vax1* transcripts were detected in the OS and OD, but not in the neural retina (NR). This in situ hybridization signal was absent in *Vax1^lacZ/lacZ* homozygous knock-in mouse eyes (bottom). (**B**) The distribution of Vax1 protein in the NR (i and iii) and OS (ii and iv) of E14.5 *Vax1^lacZ/+* and *Vax1^lacZ/lacZ* embryos was compared with that of β-gal expressed from the *lacZ* gene at the *Vax1* locus by co-staining with rabbit anti-Vax1 (green) and mouse anti-β-gal (red) antibodies. Vax1 protein detected in *Vax1^lacZ/+* mouse retinal cells, where β-gal signals were absent, is presumed to originate from external sources that co-express Vax1 and β-gal. Red dots in (iii) are non-specific background β-gal immunostaining. (**C**) Distribution of Vax1 in RGC axons and cell bodies was examined by co-immunostaining for Vax1 (green) and the RGC axonal marker NF160 (red). Nuclei were counterstained with DAPI (blue). Arrowheads in (ii) indicate Vax1 protein that co-localizes with NF160, whereas arrows point to Vax1 in APC nuclei. Vax1 immunostaining signals were completely absent in the OS and NR of *Vax1^lacZ/lacZ* mice, whereas NF160 immunostaining was still detectable in defasciculated RGC axons (iii and iv). (**D**) Sections of E18.5 WT and *Vax1^lacZ/lacZ* mouse retinas (top) and optic nerves (ON; bottom) were immunostained with rabbit anti-Vax1 antibody and gold (25 nm)-labeled anti-rabbit IgG. Subcellular localization of Vax1 proteins was then examined by electron microscopy. Arrowheads in RGC images point to Vax1 proteins in the intracellular vesicle, whereas arrows in the images indicate Vax1 proteins bound to the extracellular surface of the RGC plasma membrane (top). Arrowheads in APC images indicate Vax1 proteins in trafficking vesicles, whereas arrows mark Vax1 proteins associated with chromatin in the nucleus (bottom). Scale bars in (**A**) to (**C**): 200 μm (left column) and 20 μm (right two columns). Scale bars in (**D**): 0.5 μm (left column) and 0.2 μm (right two columns).

The following figure supplements are available for figure 4:

**Figure supplement 1**. Expression of Vax1 mRNA and protein in the vHT area.

**Figure supplement 2**. Cytoplasmic localization of Vax1 in RGCs.

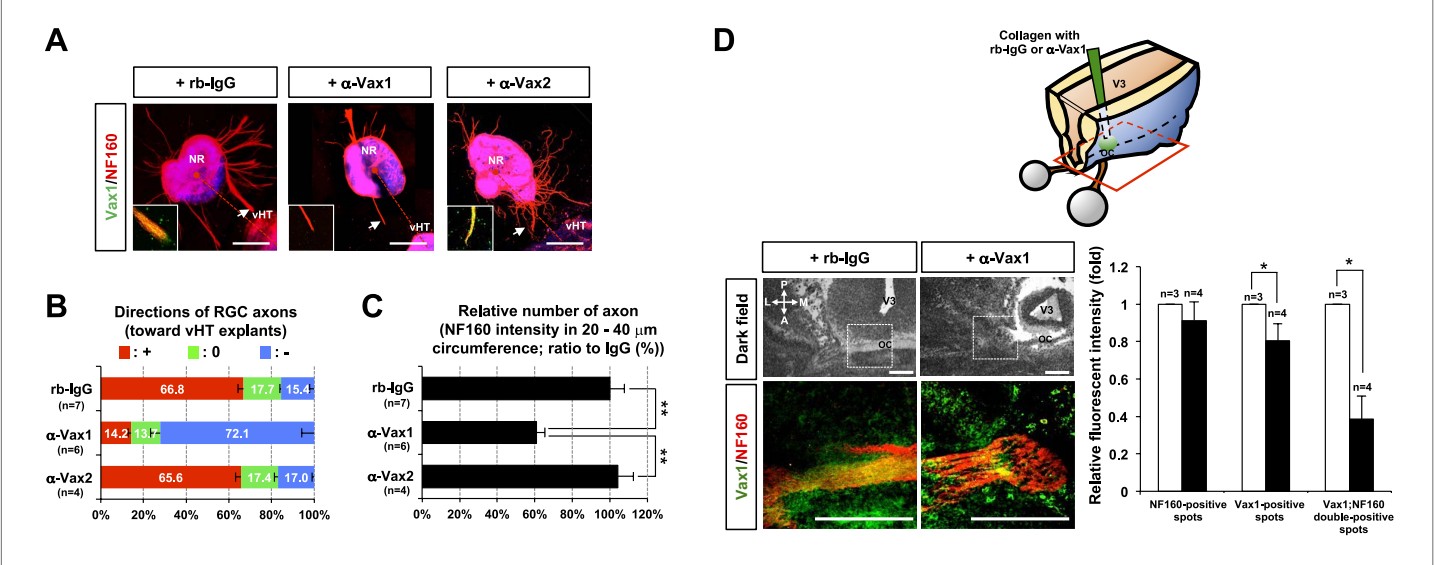

**Figure 5**. Secreted Vax1 protein is necessary for RGC axon growth. (**A**) vHTs and retinas isolated from E13.5 mouse embryos were co-cultured for 48 hr in the presence of pre-immune rabbit IgG (rb-IgG; 1 μg/ml), anti-Vax1 (α-Vax1; 1 μg/ml), or anti-Vax2 (α-Vax2; 1 μg/ml) antibodies. Vax1 localization in RGC axons was then determined by staining explants with rabbit α-Vax1 (green) and mouse α-NF160 (red). Arrowheads indicate the area magnified in each inset. Scale bars: 500 μm. (**B**) The distribution of RGC axons in each angle segment was determined as described in 'Materials and methods'. The values in the bar are averages, and error bars denote SDs. p-values are between 0.01 and 0.005 (ANOVA). (**C**) Total image pixel counts of NF160 immunofluorescence in a 20–40-μm area were compared to obtain the relative number of axons projected from each explant. Scores under y-axis labels of (**B**) and (**C**) are the numbers of explants analyzed in three independent experiments (**p < 0.001; ANOVA). (**D**) Slabs of mouse heads including eyes, forebrain, and midbrain structures were prepared from E13.5 WT mouse embryos. The third brain ventricles of mouse-head slabs were then implanted with collagen gels containing rb-IgG (1 μg/ml) or α-Vax1 (1 μg/ml) and subsequently incubated at 37°C in a $CO_2$ incubator for 12 hr (top row; see 'Materials and methods' for details). The slabs were then fixed and frozen to obtain horizontal sections (18 μm thick). The slides containing optic nerves (ON) were then further co-stained with α-Vax1 (green) and α-NF160 (red) and analyzed using an Olympus FV1000 confocal microscope. Images in the bottom row are magnifications of dotted boxes in the top row. Scale bars: 200 μm. Relative fluorescence intensities of Vax1- and/or NF160-positive immunostaining intensities in the midline area (dotted box) were measured using ImageJ software and presented graphically. White column, rb-IgG; black column, α-Vax1. The values are relative intensities compared with rb-IgG-treated samples; error bars denote SDs and values on the top of graph columns are number of slabs analyzed (*p < 0.01; Student t test). A, anterior; P, posterior; M, medial; L, lateral; *, optic chiasm; V3, third ventricle.

2 (Otx2) (*Joliot et al., 1998*; *Sugiyama et al., 2008*; *Spatazza et al., 2013*). However, little is known about the regulatory mechanisms underlying the trafficking of homeodomain transcription factors. We therefore searched for the gene encoding proteins capable of modifying the intercellular transfer of Vax1 in *Drosophila* (**Figure 6**; *screening results are unpublished*). One of the genes isolated in this screen encodes the transmembrane heparan sulfate proteoglycan (HSPG) protein, syndecan (Sdc) (*Spring et al., 1994*). HSPGs, including Sdc2, Sdc3, and glypican 1 (Glp1), are highly expressed in mouse RGC axons and have been proposed to play critical roles in RGC axon guidance in various vertebrates (*Chung et al., 2001*; *Inatani et al., 2003*; *Lee and Chien, 2004*; *Pratt et al., 2006*). We thus focused on the potential role of HSPGs as receptors for Vax1 in RGC axons.

Vax1-GFP protein was co-expressed with DsRed protein in the A/P (anterior/posterior) boundary cells of *Drosophila* wing imaginal disc under the control of a *Ptc-Gal4* driver. Vax1-EGFP, but not DsRed, was transferred to neighboring cells, results similar to those observed in the mammalian systems (**Figure 6**, top row). However, the transfer of Vax1-EGFP to neighboring wing disc cells was suppressed upon co-expression of Sdc in the A/P boundary cells (**Figure 6**, third row). Overexpressed dally-like protein (Dlp), a *Drosophila* homolog of Glp, also suppressed Vax1 transfer in the same manner as Sdc (**Figure 6**, bottom row), suggesting that the intercellular transfer of Vax1 in *Drosophila* wing imaginal disc cells is mediated by HSPGs but not specifically by Sdc. These results also imply that these overexpressed HSPGs in the A/P boundary cells captured co-expressed Vax1-EGFP protein, thereby interfering with the transfer of Vax1-EGFP to neighboring cells (**Figure 6**, diagram in the right column). Conversely, Vax1-EGFP proteins were transferred farther in the *Sdc* mutant (*Sdc23*) wing imaginal disc, where total HSPG levels were reduced owing to the loss of Sdc (**Figure 6**, second row).

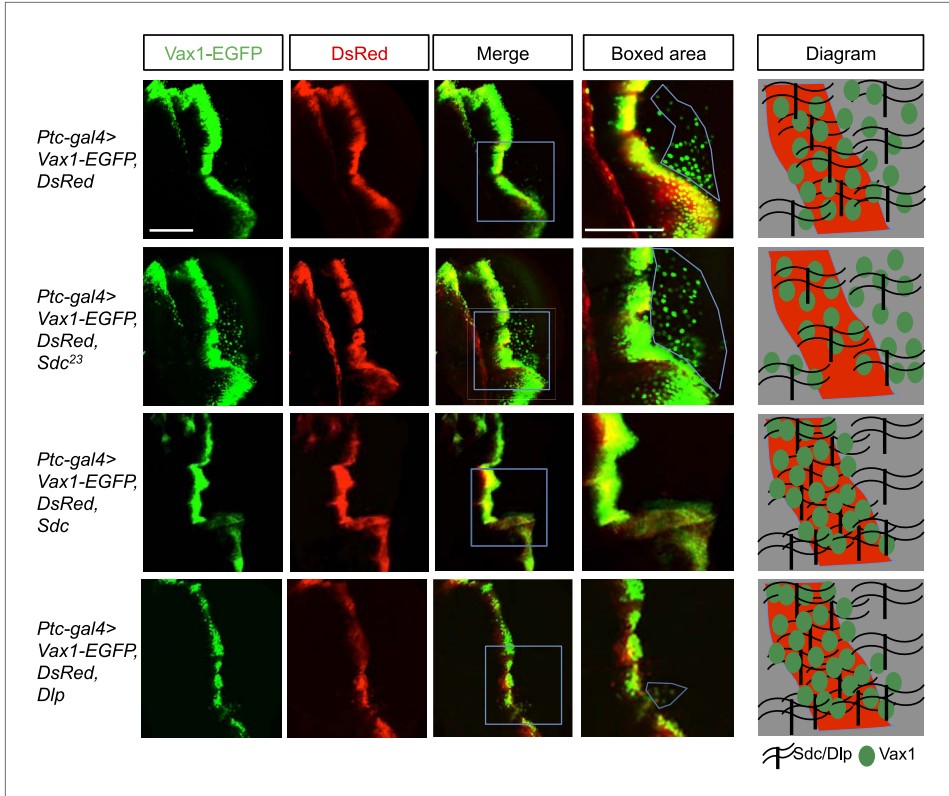

**Figure 6**. Regulation of intercellular Vax1 transfer by HSPGs in *Drosophila* wing imaginal discs. Vax1-EGFP (green) and DsRed (red) were co-expressed under the control of *Ptc-Gal4* in the A/P boundary cells of *Drosophila* wing imaginal discs in *wt* (top row) and *Sdc* mutant (*Sdc²³*; second row) flies. The number of cells positive for Vax1-EGFP but negative for DsRed at the posterior part of the wing disc was increased in *Sdc²³* flies. In contrast, the number of Vax1-EGFP-positive cells in the posterior wing disc was significantly decreased in fly embryos that co-expressed Sdc (third row) or Dlp (bottom row) together with Vax1-EGFP and DsRed. Diagrams in the right column show the distribution of Vax1-EGFP-positive cells (green) and DsRed (red) in the corresponding fly wing discs. Scale bars: 100 μm.

## Heparan sulfate-dependent binding of Vax1 to HSPGs is required for Vax1-induced RGC axonal growth

HSPG-regulated Vax1 transfer was further investigated in the mammalian systems. We found that Vax1, but not Vax2, interacted with Sdc2 in E14.5 mouse optic nerves as well as with overexpressed Sdc1 and Sdc2 in human embryonic kidney (HEK) 293T cells (*Figure 7A*, *Figure 7—figure supplement 1A,B*). Sdc2-N, lacking the C-terminal cytoplasmic domain, was able to interact with Vax1, whereas Sdc2-C, which lacks the N-terminal extracellular domain, failed to interact with Vax1 (*Figure 7—figure supplement 1C*), suggesting that Vax1 binds to the extracellular domain of Sdc.

The extracellular domain of Sdc is attached by heparan sulfate (HS) sugars (*Bishop et al., 2007*); thus, Vax1 could interact with the sugar groups as well as the protein backbone of Sdc. To determine the potential binding of Vax1 to HS side chains of Sdc2 in RGC axons, we used co-immunoprecipitation assays to test whether excess free heparin competed with HS side chains of HSPGs, including Sdc2, Sdc3, and Glp1, for binding to Vax1. Vax1 interactions with each of these HSPGs expressed in RGC axons in E14.5 optic nerves were disrupted in the presence of free heparin, whereas interactions with Pax2 (paired homeobox 2), which complexes with Vax1 in OS APCs, were not (*Figure 7B*). Moreover, recombinant Vax1-His specifically bound HS-sepharose resin with high affinity (*Figure 7C*). Collectively, these results suggest that Vax1 preferentially binds to HS side chains of HSPG proteins expressed in RGC axons.

We also examined the influence of Vax1 binding to HSPGs on RGC axon growth. Sdc3 was expressed in E14.5 mouse RGCs in a non-polarized manner and co-localized with Vax1 in RGC axons (*Figure 7—figure supplement 2*). The growth stimulatory effects of Vax1 on RGC axons were abolished by the

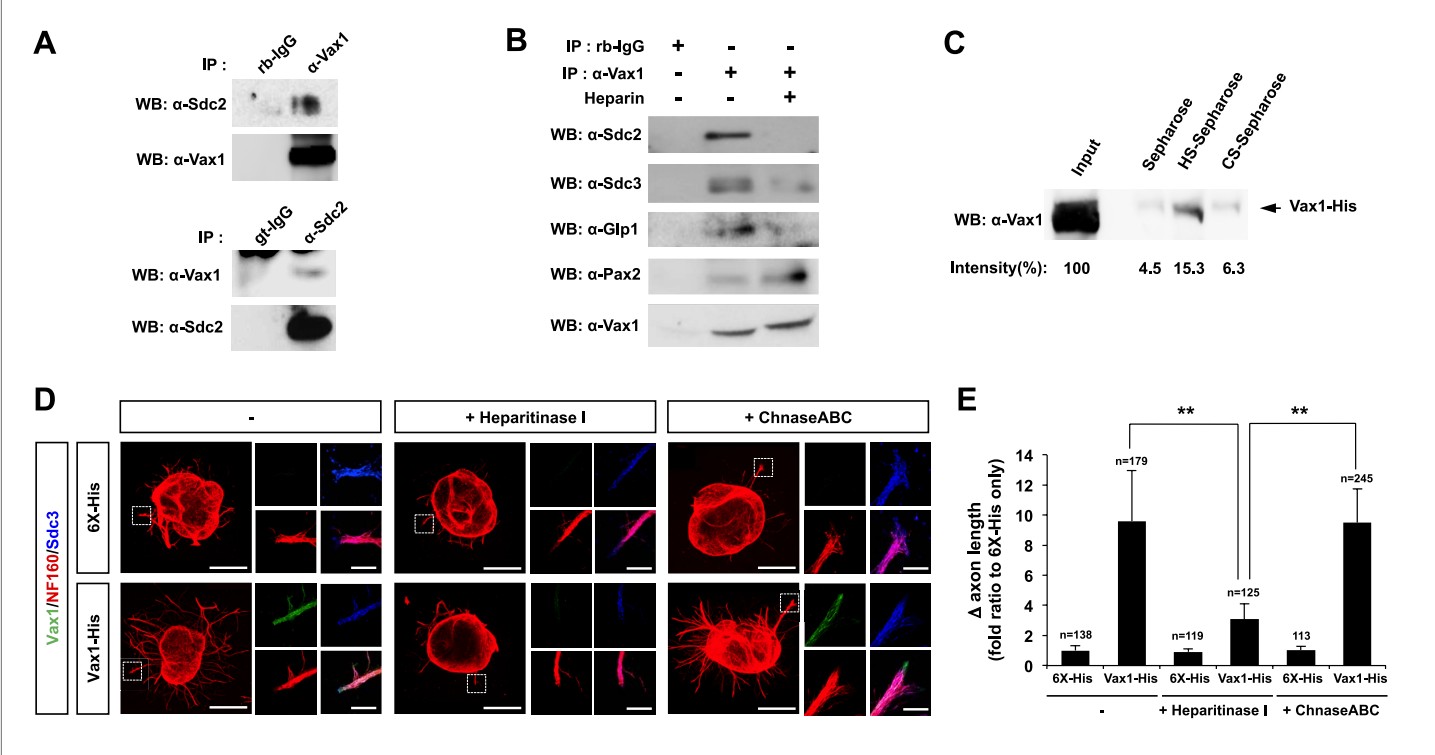

**Figure 7**. Vax1 binding to HSPGs is necessary for RGC axonal growth. (**A**) Interaction between Vax1 and Sdc2 in the E14.5 mouse optic nerve was investigated by immunoprecipitation with a rabbit anti-Vax1 (top) or goat anti-Sdc2 (bottom) antibody and subsequent Western blotting with reciprocal antibodies. The specificity of anti-Vax1 and anti-Sdc2 antibodies was confirmed by immunoprecipitation with pre-immune rabbit IgG (rb-IgG) and goat IgG (gt-IgG), respectively. (**B**) Immunoprecipitation of Vax1 in the E14.5 mouse OS was also performed in the presence or absence of heparin (1 mg/ml) to determine whether the Vax1 protein bound to HS sugar groups of Sdc2, Sdc3, and/or Glp1 HSPGs expressed in RGC axons. (**C**) Vax1-His protein (final concentration, 2 μg/ml) was incubated at 4°C for 1 hr with Sepharose 4B resin (Sigma, St. Louis, MO, USA) coated with HS or CS. The resins were washed three times with PBS, and Vax1 protein bound to the resins was eluted in SDS sample buffer for subsequent SDS-PAGE on 10% gels and Western blotting with α-Vax1. Relative intensities of Vax1 bands in Western blot images were analyzed using ImageJ software. (**D**) Retinal explants were treated with heparinase I (2.5 U/ml) or ChnaseABC (2.5 U/ml) for 3 hr and then incubated with 6X-His peptide (25 ng/ml) or Vax1-His recombinant protein (100 ng/ml) for an additional 24 hr. The presence of Vax1-His in RGC axons was then examined by co-immunostaining with rabbit anti-Vax1 (green), mouse anti-NF160 (red), and goat anti-Sdc3 (blue) antibodies. Dotted boxes indicate the area magnified at right. Scale bars: 500 μm. (**E**) The graph shows relative distances that RGC axons grew during the 24-hr incubation period. The values in the graph are averages of fold ratios compared with those of 6X-His-treated samples, error bars denote SDs, and the scores on top of graph columns are the number of axons analyzed (*p < 0.01, **p < 0.001; ANOVA). Results were obtained from three independent experiments. Numbers of explants analyzed: 6X-His, n = 6; Vax1, n = 7; heparinase, n = 5; chondroitinase, n = 5; Vax1+heparinase I, n = 6; Vax1+ChnaseABC, n = 6.

The following figure supplements are available for figure 7:

**Figure supplement 1**. Molecular determination of the interaction between Vax1 and Sdc.

**Figure supplement 2**. Co-localization of Vax1 and Sdc3 in RGC axons.

treatment of retinal explants with heparinase I, which cleaves heparin and HS sugar chains, but not by incubation with chondroitinase ABC (ChnaseABC), which digests chondroitin sugar chains (*Bishop et al., 2007*; *Figure 7D,E*). Heparinase I treatment also decreased the immunostaining intensity of exogenously provided Vax1-His in RGC axons (*Figure 7D*). Neither heparinase I nor ChnaseABC influenced RGC axon growth in the absence of Vax1 (*Figure 7D,E*). Collectively, these results suggest that the binding of Vax1 to HSPGs is necessary for the induction of RGC axonal growth.

## Vax1-induced RGC growth requires local protein synthesis

Secreted Vax1 not only bound to HSPGs at the RGC cell surface, it also moved into the RGC axoplasm by exploiting the cell-penetrating property of its homeodomain (*Joliot and Prochiantz, 2004*;

*Figure 4D*; *Figure 4—figure supplement 2*). It is therefore possible that Vax1 stimulates RGC axonal growth either by acting as a ligand for HSPGs or by regulating cytoplasmic events after penetration. To answer this question, we tested the function of a Vax1(WF/SR) mutant, in which conserved Trp147 and Phe148 amino acids responsible for cell penetration were replaced with Ser167 and Arg148 (*Joliot et al., 1998*); this mutant lacks the ability to cross the cell membrane but remains capable of binding to Sdc2 (*Figure 8—figure supplement 1A,B*). We found that Vax1(WF/SR) barely penetrated RGC axons and induced RGC axonal growth less efficiently than WT Vax1 (*Figure 8—figure supplement 1C,D*). These results suggest that Vax1-induced RGC axonal growth requires cell penetration.

To determine which cytoplasmic events Vax1 might affect, we identified cytoplasmic Vax1-interacting proteins by MALDI-TOF (matrix-assisted laser desorption/ionization-time of flight) mass spectrometry (*Figure 8A*). Interestingly, a majority of proteins isolated by Vax1-affinity purification were related to protein synthesis, including ribosomal proteins (RPs) L11, L23A, L26, S14, and S16; translation regulators, such as eukaryotic translation initiation factor (eIF) 3B and 3C; and the chaperone HSPA1A (heat shock 70-kDa protein 1A). These data suggest that Vax1 might act in RGC axons by modulating protein synthesis, a mechanism similar to that by which cytoplasmic En2 controls RGC axonal growth (*Brunet et al., 2005*; *Yoon et al., 2012*). To confirm this, we tested the effects of Vax1 on protein synthesis in RGC axons by quantifying the newly synthesized proteins, measuring the fluorescence intensity of incorporated bioorthogonal noncanonical amino acid azidohomoalanine (AHA), labeled with Alexa Fluor 488 by click chemistry (*Dieterich et al., 2007*). Treatment with Vax1-His induced a remarkable increase in the fluorescence intensities of AHA-Fluor 488-labeled proteins in RGC axons (*Figure 8B*, middle column), whereas the Vax1(WF/SR)-His had no effect on protein synthesis (*Figure 8B*, right column), indicating that intracellular Vax1 stimulated protein synthesis.

In contrast to its significant induction of protein synthesis in axons, Vax1-His-induced effects on AHA-Fluor 488 fluorescence intensities in retinal cell bodies were not notably different from those of Vax1(WF/SR)-His or the 6X-His (*Figure 8B*, bottom row). Similarly, Vax1 failed to stimulate general translation in the cultured cell-lines and purified polysomes in vitro (data not shown). In contrast, isolated retinal axons from the cell body still responded to Vax1 as efficiently as those that projected from intact retinal explants (*Figure 8C,D*). Taken together, these results suggest that Vax1 stimulates RGC axon growth by stimulating local translation of specific mRNA in the axon rather than by modulating gene expression in the cell body.

## Imported Vax1 promotes the access of RGC axons to the midline

We further investigated whether extracellular Vax1 could restore RGC axon growth towards the midline in *Vax1⁻/⁻* mice by re-supplying Vax1 protein to the vHT extracellular space. In contrast to the lack of RGC axon access to the vHT observed in *Vax1⁻/⁻* mouse embryos, remarkable numbers of RGC axons were detectable in the vHT of *Vax1⁻/⁻* mouse embryos implanted with collagen gels that released recombinant Vax1-His (*Figure 9A*, top rows of left two columns, B). Remarkable number of RGC axons were able to grow to the source of extracellular Vax1 protein (i.e., the third ventricle), although they failed to restore the OC. In contrast, implantation of collagen gels that released cell-penetration–defective Vax1(WF/SR)-His did not induce the regrowth of RGC axons (*Figure 9A, [third column], B*). Implanted recombinant Vax1-His was detectable in RGC axons, whereas Vax1(WF/SR)-His was not, suggesting that the implanted Vax1 stimulated RGC axon growth by penetrating into the axoplasm.

It has been suggested that the avoidance of RGC axons in *Vax1⁻/⁻* mouse vHT might result from high concentrations of Slit protein in the ventral-lateral diencephalon (*Bertuzzi et al., 1999*). In support of this, the growth of RGC axons towards *Vax1⁻/⁻* vHT explants were partly recovered by sequestering extracellular Slit protein using an Fc-fused extracellular fragment of the Slit receptor Robo1 (Robo1-Fc) (*Figure 9—figure supplement 1*). Thus, we further investigated the roles of Slit in RGC axon avoidance in *Vax1⁻/⁻* mouse embryos using Robo1-Fc. The growth of RGC axons into the vHT was partially rescued by implanting a Robo1-Fc–releasing collagen gel into the third ventricle, an effect that was further enhanced by implantation of gels co-releasing Vax1-His and Robo1-Fc (*Figure 9A*, two right columns, B).

## Discussion

RGC axonal projection to the OC is often compared to the spinal commissural axonal projection toward the floor plate (FP). Spinal commissural axons are prevented from prematurely entering the

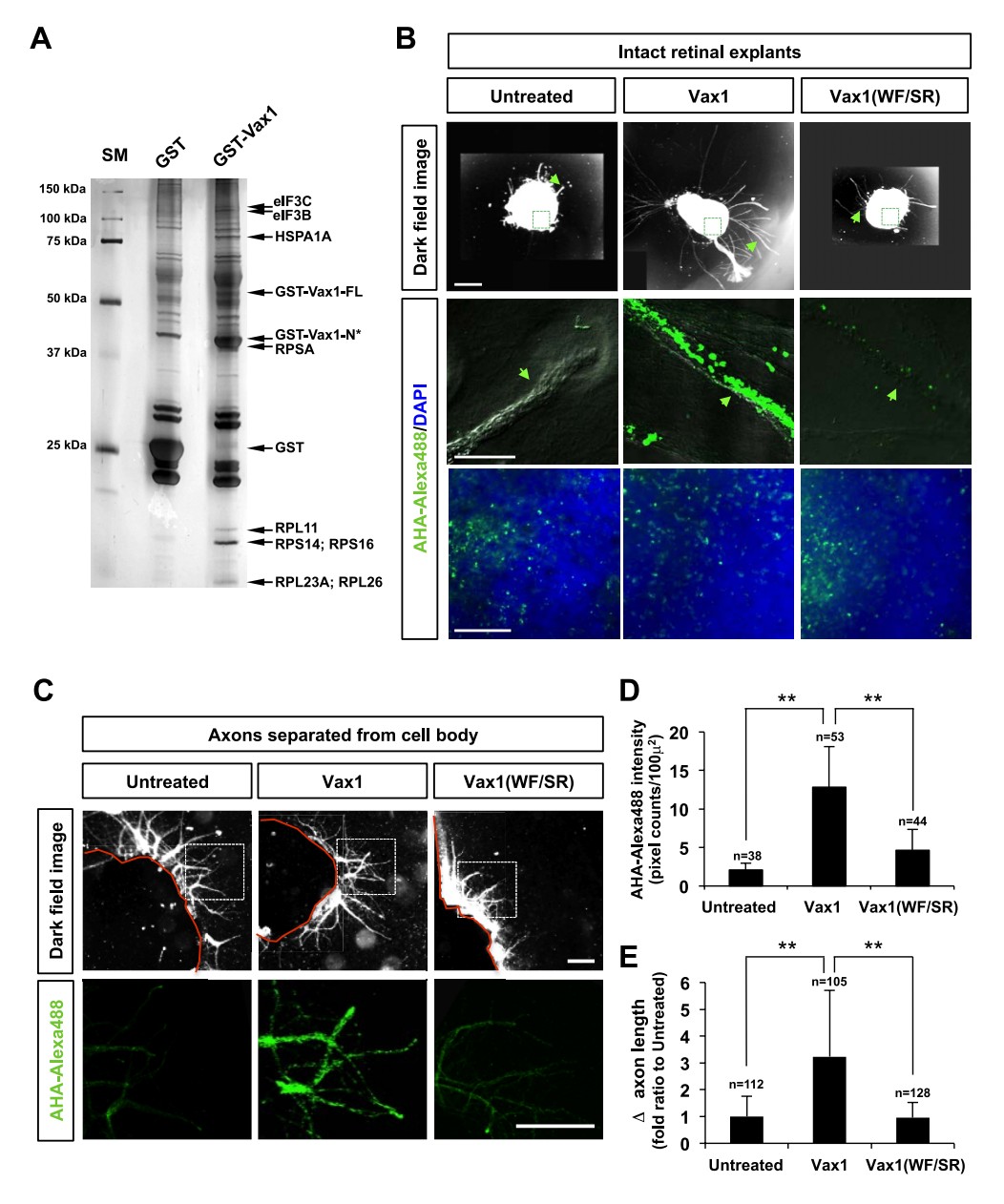

**Figure 8**. Imported Vax1 induces RGC axonal growth by stimulating local protein synthesis. (**A**) GST and GST-Vax1 protein complexes were affinity purified from cytoplasmic fractions of HEK293T cells overexpressing GST and GST-Vax1, respectively (see 'Materials and methods' for details). Complexes were then analyzed by SDS-PAGE on 10% gels and subsequent silver staining to detect proteins specifically enriched in GST-Vax1 complexes. The identities of protein bands, shown to the right of the gel photograph, were determined by MALDI-TOF mass spectrometry. Vax1-FL, full-length Vax1; Vax1-N*, Vax1 N-terminal fragment. (**B**) E13.5 mouse retinal explants were cultured for 24 hr before changing to medium containing Vax1-His (100 ng/ml) or Vax1(WF/SR)-His (100 ng/ml) for an additional 16 hr-incubation. The explants were further incubated for 6 hr after addition of the bioorthogonal noncanonical amino acid AHA (L-azidohomoalanine). Newly synthesized proteins incorporating these noncanonical amino acids were labeled with Alexa Fluor 488-alkyne by click chemistry (*Dieterich et al., 2007*), and the rates of protein synthesis in RGC axons (middle row) and explant cell body (bottom) were assessed by measuring the fluorescence intensities of AHA-Alexa Fluor 488-labeled proteins (see 'Materials and methods' for details). Scale bars: 500 μm (top) and 100 μm (bottom). (**C**) The influence of nuclear events in Vax1-induced RGC axon growth was excluded by isolating axons from the cell body before treatment with Vax1 proteins (100 ng/ml) for 6 hr. Arrowheads indicate the area magnified in each inset. Scale bars: 500 μm. (**D**) Relative AHA-Alexa488 fluorescence intensities in

*Figure 8. Continued on next page*

*Figure 8. Continued*
cell body-free axons were measured using ImageJ software and are shown graphically. Error bars denote SDs.
(**E**) Relative distances that RGC axons grew during this 6-hr incubation period are presented graphically. The values
in the graph are averages of fold ratios compared with those of untreated samples. Scores on top of the graph
columns in (**D**) and (**E**) are number of axons analyzed (**p < 0.001; ANOVA). Results were obtained from two
independent experiments. Numbers of explants analyzed: untreated, n = 4; Vax1, n = 5; Vax1(WF/SR), n = 4.
The following figure supplement is available for figure 8:
**Figure supplement 1**. Biophysical properties of Vax1(WF/SR) mutant protein.

midline by Slit1 expressed in the medial spinal cord and grow in the ventral direction (***Stein and Tessier-Lavigne, 2001***). In much the same way, Slit1 in the preoptic area and ventral-lateral diencephalon prevents RGC axons from accessing the brain anywhere but at the vHT to form the OC (***Erskine et al., 2000***; ***Plump et al., 2002***). The spinal commissural axons sense attractive cues, such as netrin and Shh, secreted from the FP (***Stein and Tessier-Lavigne, 2001***; ***Charron et al., 2003***). The attractive netrin and Shh signals are expected to compete with the co-existing repulsive signal of Slit2, which can be accumulated locally by HSPGs, including α-dystroglycan, at the ventral FP (vFP), to determine the directionality of spinal commissural axon growth cones (***Matsumoto et al., 2007***; ***Wright et al., 2012***). RGC-expressed HSPGs were also reported to play roles as co-receptors for netrin and Slit (***Johnson et al., 2004***; ***Hussain et al., 2006***; ***Piper et al., 2006***; ***Ogata-Iwao et al., 2011***), suggesting a similar HSPG-based antagonistic regulation of RGC axon growth. However, both netrin and Shh are dispensable with respect to attracting RGC axons toward the vHT and instead function as repulsive cues for RGC axons (***Deiner and Sretavan, 1999***; ***Sanchez-Camacho and Bovolenta, 2008***). In this study, we propose vHT-secreted Vax1 as a RGC axon growth factor that is analogous to vFP-secreted netrin and Shh. This unconventional axon growth factor also utilizes HSPGs for anchoring to RGC axons (***Figure 7***) and

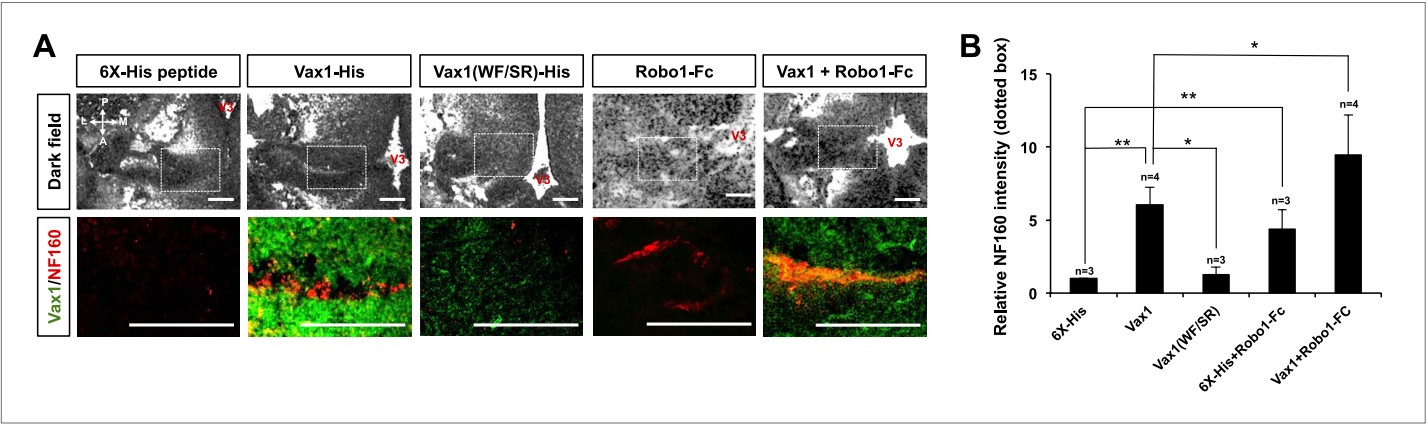

**Figure 9**. RGC axons re-grow in Vax1-implanted, *Vax1⁻/⁻* mouse brains. (**A**) The third ventricles of E13.5 *Vax1⁻/⁻* mouse-head slabs were implanted with collagen gels mixed with 6X-His peptide (4.78 μg/ml; 5.69 nmol), Vax1-His (200 μg/ml; 5.69 nmol), or Vax1(WF/SR)-His (200 μg/ml; 5.69 nmol) and incubated for 12 hr (see diagram in ***Figure 5D*** and 'Materials and methods' for details). *Vax1⁻/⁻* mouse-head slabs were also implanted with collagen gels mixed with Robo1-Fc fragment (1 μg/ml; 12.35 pmol; R&D Systems, Minneapolis, MN, USA) in the presence of 6X-His peptide (4.78 μg/ml; 5.69 nmol) or Vax1-His (200 μg/ml; 5.69 nmol) and incubated for 12 hr. The fluorescence images of horizontal sections of head slabs were obtained using an Olympus FV1000 confocal microscopy equipped with a transmitted light detector (top row). The same embryonic sections were further stained with rabbit anti-Vax1 (green) and mouse anti-NF160 (red) antibodies (bottom row). (**B**) Fluorescence intensities of NF160 immunostains in the boxed areas in (**A**) were measured using ImageJ software and are presented graphically. The values are intensities expressed relative to rb-IgG-treated samples, and error bars denote SDs (**p < 0.001; ANOVA). Numbers on top of the graph columns are head-slab preparations.
The following figure supplements are available for figure 9:

**Figure supplement 1**. vHT-secreted Silt inhibits RGC axon growth.

**Figure supplement 2**. Reciprocal antagonism between Vax1 and Slit2 in vitro.

could compete with Slit for HSPGs binding in vitro (*Figure 9—figure supplement 2*). However, it is unclear whether this competition is valid in physiological conditions because Slit does not inhibit RGC axon growth toward the vHT midline (*Erskine et al., 2000*; *Plump et al., 2002*). Moreover, cell penetration-defective Vax1(WF/SR) mutant could not induce RGC axon growth in vivo as well as in vitro, despite of its capability of HSPG binding (*Figures 8 and 9*, *Figure 8—figure supplement 1*). It suggests that HSPG binding of Vax1 is not sufficient to induce RGC axon growth but local protein synthesis induced by internalized Vax1 is necessary. Collectively, we propose a hypothetical model that Vax1 promotes RGC axon growth towards the vHT midline by directly targeting mRNA in the axons rather than by serving for a conventional axon guidance molecule that binds specific receptors and triggers on intracellular signaling cascades (*Figure 10*). Identification of axonal target mRNA awaits future investigations.

Translation-dependent, but transcription-independent, RGC axon guidance by a secreted transcription factor has also been reported for En2, which regulates RGC growth cone turning by increasing the expression of mitochondrial proteins involved in elevating the local ATP, which potentiates ephrin A5 signaling (*Brunet et al., 2005*; *Stettler et al., 2012*; *Yoon et al., 2012*). Since Vax1 and En2 share a homologous homeodomain, Vax1 could function in a similar manner by cooperating with attractive RGC axon guidance cues, such as VEGF164 and NrCAM (*Williams et al., 2006*; *Erskine et al., 2011*; *Kuwajima et al., 2012*), and by modulating mitochondrial activity. Conversely, En2 could also use HSPGs to bind target axons and fine-tune their selective growth.

In their use of HSPGs as docking sites for RGC axons, Vax1 can be compared to Otx2 that binds specifically to CSPGs of the perineuronal net surrounding parvalbumin (PV) neurons in the visual cortex (*Beurdeley et al., 2012*; *Miyata et al., 2012*). The differential affinities of Vax1 and Otx2 for HS and CS might be related to their different homeodomains. Among homeodomain proteins proven to exhibit transfer, Vax1 possesses an antennapedia class homeodomain homologous to that of Emx2 and En2, whereas Otx2 shares a paired class homeodomain similar to that of Pax6 (paired box 6) (*Bürglin, 2011*;

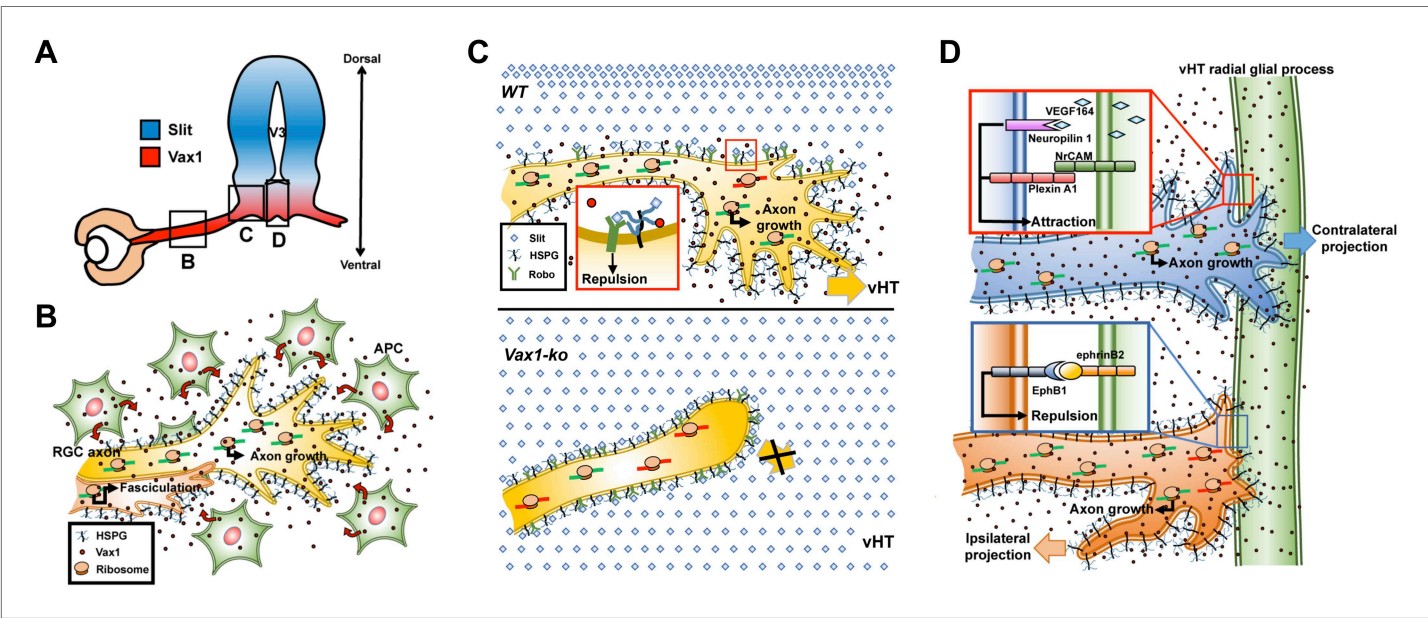

**Figure 10**. Model depicting Vax1 functions as a secreted retinal axon growth factor. Vax1 is expressed in radial glia and NPCs of the vHT as well as the OS APCs (**A**) and secreted to the extracellular space (**B**). RGC axons that grow in the OS capture APC-secreted Vax1 by HSPGs, resulting in an increase in Vax1 concentration at RGC axons for subsequent penetration and local activation of translation in the axon (**B**). The imported Vax1 in RGC axoplasm not only promotes axonal growth towards the vHT, but it also enhances fasciculation of RGC axons. This axon growth stimulatory activity of Vax1 supports sustained RGC axon growth to the vHT midline after the axons were avoided from progressing to dorsal diencephalon, which expresses high concentration of Slit repulsive axon guidance cue (**C**, top). Therefore, RGC axons stop at the lateral wall of *Vax1-ko* mouse diencephalon and fail to access the midline (**C**, bottom). At the vHT midline, RGC axon guidance cues, including VEGF164, NrCAM, and ephrinB2, determine the directionalities of RGC axon growth cones by acting their specific receptors (**D**). Vax1 does not likely determine the directionalities of RGC axon growth cone at the midline but does promote the growth of the RGC axon shaft as well as the growth cone regardless of their original positions in the retina.

*Spatazza et al., 2013*). One intriguing possibility that has not yet been explored is that these secreted homeodomain proteins share the property of preferential binding to HS and CS; however, it is at least as likely that intercellular transfer of homeodomain proteins are target-selective events.

Vax2 was not as effectively secreted from COS7 cells as Vax1 (*Figure 2*), despite sharing an identical homeodomain with Vax1. This suggests that secretion of Vax1 is not solely mediated by the homeodomain but is also dependent on a three-dimensional structure that supports the secretion property of the homeodomain. Unlike Vax1, Vax2 undergoes a specific phosphorylation that results in its cytoplasmic retention (*Kim and Lemke, 2006*). Phosphorylation of En2 inhibits its secretion (*Maizel et al., 2002*), suggesting that the phosphorylation might change the structure of these proteins in such a way that it interferes with homeodomain recognition by secretion regulators. However, phosphorylation-defective Vax1(S170A) was still unable to be secreted from COS7 cells (data not shown). Instead, Vax2 could be secreted from different types of cells, where Vax2 might form three-dimensional structures that can be recognizable by secretion machineries (*Lee et al., unpublished data*). The results suggest that the secretion of homeodomain proteins is a cell context-dependent event.

Multiple midline-crossing defects, including agenesis of the corpus callosum, anterior commissure, and hippocampal commissure, are observed in *Vax1*-deficient mice and homozygous *VAX1* mutant human patients (*Bertuzzi et al., 1999*; *Slavotinek et al., 2012*). However, the molecular functions of Vax1 in the development of these structures remain unknown. In this study, we show that Vax1-induced RGC axonal growth is independent of its transcription factor activity (*Figures 2A and 3C*). Instead, Vax1 acts as a regulator of translation after penetrating into RGC axons (*Figure 8*). Although we cannot rule out Vax1 functions as a transcription factor in the development of those commissures, our results suggest a potential role of secreted Vax1 in the growth of cortical axons (*Min et al., unpublished data*). Whether axonal growth of these commissural axons in the mammalian forebrain also requires local protein synthesis triggered by Vax1 protein secreted from cells located in other midline structures, such as the ventral-medial telencephalon and the septum, is a question that warrants further investigation.

## Materials and methods

### Mice and explant culture

*Vax1* knock-out (*Vax1*$^{-/-}$) and *Vax1* knock-in (*Vax1*$^{lacZ/lacZ}$) mice were reported previously (*Bertuzzi et al., 1999*; *Hallonet et al., 1999*). Retinal and vHT explants were cultured as described previously (*Sato et al., 1994*). Briefly, retinal or vHT explants were added to a collagen mixture and positioned on plates coated with poly-L-lysine (10 µg/ml) and laminin (10 µg/ml). The explants were then incubated at 37°C for 1 hr to allow gelling before adding Neurobasal medium containing B27 supplement (Invitrogen, Carlsbad, CA, USA). COS7 cell droplets (10$^5$ cells/droplet) were also prepared using the same procedures. The explants were cultured alone or co-cultured with vHT or COS7 explants for 48 hr before treating with proteins and antibodies.

For time-lapse recording of retinal axon growth, mouse retinal explants were treated with FITC-labeled 6X-His peptide (100 ng/ml, 500 ng/ml) or Vax1-His protein (500 ng/ml) for 24 hr and photographed every 15 min using a Zeiss Axio Observer Z1 inverted microscope. The explants were washed twice with phosphate-buffered saline (PBS) prior to incubation with rabbit anti-Vax1 polyclonal antibody (α-Vax1; 1:100) for 30 min to detect Vax1-His located on the cell surface. The explants were then washed with PBS and fixed in 4% paraformaldehyde (PFA)/PBS for subsequent immunostaining procedures to detect total Vax1-His protein inside and outside of cells.

For slab embryo culture with collagen gel implantation, E13.5 mouse embryos were moved onto culture slide chambers containing collagen mixture, positioning the dorsal part on top, after dissecting out the dorsal half of the brain and lower part of the mouth. Droplets of collagen solution mixed with pre-immune IgG (1 µg/ml), α-Vax1 (1 µg/ml), 6X-His peptide (4.78 µg/ml), or Vax1-His (200 µg/ml) protein were then delivered into the third ventricle of the slab embryos. The embryos were then filled with culture medium and incubated for 12 hr at 37°C in a humidified atmosphere supplemented with 7% $CO_2$. The embryos were fixed in 4% PFA/PBS for subsequent freezing in OCT (optimal cutting temperature) medium. RGC axon growth at the vHT was monitored along the horizontal axis of slide-mounted embryonic brain sections under an Olympus BX-71 microscope. The slides were then stained with appropriate antibodies and examined under an Olympus FV1000 confocal microscope to detect the penetration of implanted Vax1 protein into RGC axons.

## Retinal axon count

Relative axon counts (combined values of numbers and lengths of axons) of retinal explants were obtained by measuring NF160-fluorescent pixels in images of retinal axons using the ImageJ software. Relative axon counts of retinal explants co-cultured with the vHT explants or COS7 cell aggregates were obtained along three angle segments: forward (+), neutral (0), and reverse (−). A clockwise angle from a line connecting two centers of explants was obtained and classified as forward direction (+) if it was between 0° and 60° or between 301° and 360°; neutral direction (0) if it was between 61° and 120° or between 241° and 300°; and reverse direction (−) if it was between 121° and 240° (*Figures 2B and 5B*). Relative axon counts in each angle segment were then obtained by comparing the pixel counts of NF160 immunofluorescence in RGC axons of the explants in each angle segment.

## Antibodies

α-Vax1 and α-Vax2 were produced as reported previously (*Mui et al., 2005*). Commercially available antibodies against the following proteins were used: mouse anti-Myc (Santa Cruz Biotechnology, Dallas, TX, USA), mouse anti-GFP (Santa Cruz Biotechnology, Dallas, TX, USA), mouse anti-tubulin β-III (Tuj1; Covance, Princeton, NJ, USA), goat anti-Sox2 (Santa Cruz Biotechnology, Dallas, TX, USA), mouse anti-Nestin (RC2; Millipore, Billerica, MA, USA), mouse anti-β-galactosidase (Developmental Studies Hybridoma Bank, DSHB), mouse anti-NF160 (Developmental Studies Hybridoma Bank [DSHB], Iowa City, IA, USA), goat anti-Sdc2 (Santa Cruz Biotechnology, Dallas, TX, USA), goat anti-Sdc3 (Santa Cruz Biotechnology (Dallas, TX, USA), for immunohistochemistry), rabbit anti-Sdc3 (Abcam (UK), for Western blot), rabbit anti-Glp1 (Santa Cruz Biotechnology, Dallas, TX, USA), and rabbit anti-Pax2 (Invitrogen, Carlsbad, CA, USA) antibodies.

## Immunohistochemistry

The heads of embryonic mice were fixed in 4% PFA/PBS at 4°C for 2–16 hr, depending on the protein to be detected, and then incubated in a 20% sucrose/PBS solution at 4°C for 16 hr before embedding in OCT medium for freezing. Sections of frozen tissue were incubated for 1 hr in a blocking solution containing 0.2% Triton X-100, 5% normal donkey serum, and 2% bovine serum albumen (BSA) in PBS. Sections were first incubated with the indicated primary antibodies in blocking solution without 0.2% Triton X-100 at 4°C for 16 hr and then with the appropriate Alexa488-, Cy3-, or Cy5-conjugated secondary antibody. Immunofluorescence was subsequently analyzed using Olympus FV1000 and Zeiss LSM710 confocal microscopes.

## Isolation and detection of extracellular proteins in growth media and CSF

Growth medium or CSF from the lateral ventricle of E14.5 mice was centrifuged twice at 500×*g* for 10 min and then twice at 2000×*g* for 15 min to obtain supernatant (S3) fractions. The S3 fractions were then mixed with an equal volume of 3 M trichloroacetic acid (TCA) solution to precipitate the macromolecules. The TCA precipitates were washed twice with 100% acetone, air-dried pellets, and dissolved in 2×-sodium dodecyl sulfate (SDS) sample buffer for SDS-PAGE (polyacrylamide gel electrophoresis) analysis.

## *Drosophila* lines and whole-mount immunostaining

*Ptc-Gal4, UAS-DsRed, UAS-Sdc, UAS-Dlp*, and *sdc²³* flies were obtained from the Bloomington stock center, IN, USA. The UAS-*Vax1-EGFP* fly was generated by injection of pUAS-Vax1-EGFP constructed by cloning mouse Vax1 cDNA into the pUAST-EGFP vector. Third-instar larvae of *Ptc-Gal4>UAS-Vax1-EGFP,UAS-DsRed;+* were obtained from a cross of the *Ptc-Gal4>UAS-DsRed;TM6B* fly with a *UAS-Vax1-EGFP* fly.

After fixing larval wing imaginal discs with 4% PFA/PBS for 30 min, the *Ptc-Gal4*-induced green fluorescence signals from EGFP and Vax1-EGFP proteins were compared with red fluorescence signals from DsRed proteins by confocal microscopy (Olympus FV1000).

## Co-immunoprecipitation

HEK293T cells and E14.5 mouse optic nerves were lysed in a buffer consisting of 10 mM Tris–HCl (pH 7.4), 200 mM NaCl, 1% Triton X-100, and 1% NP-40. Cell lysates were centrifuged at 12000×*g* for 10 min at 4°C. The supernatants were collected and incubated with the indicated antibodies at 4°C

for 16 hr; then, protein A-agarose beads were added and incubation was continued at 4°C for 1 hr. After washing the immune complexes five times with lysis buffer, proteins were eluted with 2× SDS sample buffer. Samples were then analyzed by SDS-PAGE and Western blotting.

## GST-affinity purification of Vax1-interacting proteins

HEK293T cells transfected with pEBG or pEBG-Vax1 were lysed with a buffer consisting of 10 mM Tris–HCl (pH 7.4), 200 mM NaCl, and 1% NP-40. Cell lysates were centrifuged at 12000×$g$ for 10 min at 4°C. The supernatants were collected and incubated with glutathione Sepharose 4B resin (GE Healthcare) at 4°C for 1 hr. After washing five times with lysis buffer, proteins were eluted with 2× SDS sample buffer and samples were analyzed by SDS-PAGE on 10% gels. Gels were stained using Silver Stain Kit for Mass Spectrometry (Pierce) to isolate bands for MALDI-TOF mass spectrometry analysis at the Korea Basic Science Institute (KBSI), Daejeon, South Korea.

## Click chemistry for labeling newly synthesized proteins

Explant culture media were replaced with methionine-free media, 30 min prior to the addition of 50 µM L-azidohomoalanine (AHA, Invitrogen (Carlsbad, CA, USA)). After 6 hr, retinal explants were washed twice with PBS containing 1% fetal bovine serum (FBS) and then 30 µM DIBO-Alexa Fluor 488 (Invitrogen, Carlsbad, CA, USA) in PBS containing 1% FBS was added. The explants were then incubated at room temperature in the dark for 1 hr. After washing four times in PBS containing 1% FBS, retinal explants were fixed with 4% PFA in PBS for 15 min at room temperature for subsequent detection of the fluorescence of proteins incorporating AHA-Alexa Fluor 488.

## Acknowledgements

We thank Drs. Greg Lemke, Peter Gruss, and Eok-Soo Oh for providing *Vax1-ko* mice, *Vax1^lacZ* knock-in mice, and GFP-Sdc2 constructs, respectively. We also thank to Dr. Jong Soon Choi for the support with MALDI-TOF mass spectrometry analysis. This work was supported by grants from the Global Research Laboratory Program (NRF-2009-00424; JWK), Brain Research Program (NRF-2013-056566; JWK), Basic Science Research Program (NRF-2014R1A2A2A01003069; JWK), and Stem Cell Research Program (NRF-2006-2004289; KHK) funded by the Korean Ministry of Science, ICT, and Future Planning (MSIP). This study was performed in strict accordance with the recommendations in the Guide for the Care and Use of Laboratory Animals of the Korean Ministry of Agriculture, Food, and Rural Affairs. All of the animals were handled according to approved institutional animal care and use committee (IACUC) protocols (#13-130) of Korea Advanced Institute of Science and Technology.

## Additional information

### Funding

| Funder | Grant reference number | Author |
| --- | --- | --- |
| Ministry of Science, ICT, and Future Planning, Republic of Korea | NRF-2009-00424; NRF-2013-056566; NRF-2014R1A2A2A01003069 | Jin Woo Kim |
| Ministry of Science, ICT, and Future Planning, Republic of Korea | NRF-2006-2004289 | Kyung Hwa Kang |

The funder had no role in study design, data collection and interpretation, or the decision to submit the work for publication.

### Author contributions

NK, KWM, Acquisition of data, Analysis and interpretation of data, Drafting or revising the article, Contributed unpublished essential data or reagents; KHK, EJL, Acquisition of data, Analysis and interpretation of data, Contributed unpublished essential data or reagents; H-TK, KM, DL, S-HL, Acquisition of data, Analysis and interpretation of data; JC, Analysis and interpretation of data, Drafting or revising the article; JWK, Conception and design, Analysis and interpretation of data, Drafting or revising the article

## Ethics

Animal experimentation: This study was performed in strict accordance with the recommendations in the Guide for the Care and Use of Laboratory Animals of the Korean Ministry of Agriculture, Food and Rural Affairs. All of the animals were handled according to approved institutional animal care and use committee (IACUC) protocols (#13-130) of Korea Advanced Institute of Science and Technology.

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
