## [Decision Letter]

Thank you for sending your work entitled “Regulation of retinal axon growth by secreted Vax1 homeodomain protein” for consideration at *eLife*. Your article has been favorably evaluated by a Senior editor and 3 reviewers, one of whom is a member of our Board of Reviewing Editors.

The Reviewing editor, Marianne Bronner, and the other reviewers discussed their comments before we reached this decision, and the Reviewing editor has assembled the following comments to help you prepare a revised submission.

This is an interesting study that identifies a somewhat unexpected mechanism by which the transcription factor Vax1 regulates retinal ganglion cell (RGC) axon growth. The authors report that despite the lack of Vax1 mRNA expression in RGCs, they detect Vax1 protein in RGCs in vivo. The authors utilize in vitro experiments to demonstrate that Vax1 functions in a non-cell autonomous manner and, similar to other homeobox transcription factors, Vax1 can be secreted. The authors go on to show that similar to En-2, Vax1 is internalized by RGC axons, where it stimulates axonal outgrowth in a transcription-independent manner by upregulating axonal translation. The authors also perform a screen in drosophila for molecules that mediate the intercellular transfer of Vax1 and identify HSPGs as critical for Vax1 function. Finally, the authors perform in vivo rescue experiments in Vax1 knockout mice to show that wildtype Vax1 in the 3rd ventricle can increase RGC growth, whereas a mutant version of Vax1 that cannot penetrate RGC axons fails to rescue. Taken together, these data lead the authors to propose a model in which Vax1, secreted from external sources such as the radial glia in the optic chiasm, can penetrate RGCs in an HSPG dependent manner to upregulate local translation in axons to positively regulate their outgrowth.

The identification of homeodomain transcription factors as “axon guidance molecules” is interesting. The present study is not the first report of this phenomenon, but does provide a solid basis for making this claim.

Substantive concerns:

1) The references to other studies are deficient (missing or improper citation/interpretation) and the text should be edited for proper English usage. For example, the authors should cite papers regarding the idea that HSPGs are involved in axon guidance, and particularly at the optic chiasm, since this is not a new concept. For example: [8], J Comp Neurol 436: 236; [35], J Neurosci.26:6911; Conway et al., 2011, J Neurosci 31: 1955; Wilson and Stoeckli, 2013, Neuron 79:478; Sánchez-Arrones et al., 2013, J Neurosci 33:8596; Fabre et al., 2010, J Neurosci 30:266 and Yam and Charron, Curr Op Neurobiol 2013).

2) Although the authors show that the chiasm appears normal by RC2 staining, they should confirm that the expression of known chiasm localized cues (Vegf, ephrinB1, NrCAM) is normal. Similarly, there is no information on localization of HSPGs on RGCs nor is there any proposal/analysis on the fate of HSPGs after Vax1 binds: are they also internalized? If not, are they cleaved once bound to Vax1? These points should be discussed. A large number of HSPG binding proteins have been identified (e.g. FGF2, slit2, Semas, ephrins, syndecans, glypicans, etc). For all these cues HSPGs function as co-receptors for a transmembrane spanning signaling receptor. The authors should discuss this possibility and whether there is data for co-localization of these on RGCs.

3) The explants tested were taken from dorsal retina, at E13, when a few cells project ipsilaterally but ultimately disappear. The ventrotemporal retina gives rise to the permanent ipsilateral component after E14. Have the authors analyzed expression of syndecan or glypican in the “ipsi” retina and how Vax1-HSPG interactions underlie ipsi axon midline avoidance? This could be addressed by comparing ventrotemporal versus dorsotemporal retina.

4) The authors propose that vax1 overrides Robo/slit repulsion. Is slit repulsion of cultured RGC axons reduced (blocked) in the presence of recombinant vax1? Slits are thought to inhibit RGC axons from straying (defasciculating) from the main bundles at the chiasm, in an “inhibition-surround” mechanism, not as an inhibitory molecule at the crossing point (see Erskine et al., J Neurosci 2000, 20:4975; [34] Neuron 33:219). The authors placed Vax-1-containing cells in the third ventricle above the chiasm in an isolated prep, showing that growth was enhanced, and application of Fc-Robo3 also enhanced RGC axon growth (together growth was further enhanced). While this experiment argues for Slit involvement in this preparation, it cannot relate to guidance in vivo because Robo3 is not expressed in the mouse retina (see above references and Thompson et al., 2006, J Neurosci 26:8092). Further, there should be axon extension in the control preparation in Figure 9. Finally, the model posed in Figure 9 is misleading: how would the Vax1-HSPG interaction act on Slit inhibition leading to action of molecules important for contralateral crossing (e.g., VEGF164, NrCAM, and Semaphorin6D, latter, erroneously indicated for ipsilateral growth)?

5) The authors show that in contrast to Vax1, Vax2 is not secreted and does not bind to HSPGs (Figure 2, Figure 7—figure supplement 1). This is somewhat surprising, since the domains that regulate secretion of homeodomain transcription factors are typically within the homeodomain region, which is highly conserved between Vax1 and Vax2. Have the authors attempted to identify the specific domains required for secretion and HSPG binding that differ between Vax1 and Vax2? The authors should test whether the addition of exogenous Vax2 can bind to RGCs and promote their outgrowth using assays like those used in Figure 3.

6) Immuno-EM studies show that vax1 is localized to the axonal surface of RGCs and also found in endocytosed vesicles. To participate in local translation, vax1 likely needs to be in the cytoplasm not inside vesicles?

7) In the co-culture assays in Figure 1, WT retinal explants cultured with Vax1-/- vHT not only shows no attraction, but appears to show active repulsion. This may be due to the secretion of Slit from the explants, which should be tested by examining whether addition of Robo-Fc (similar to Figure 9) inhibits this repulsion. With respect to Vax localization and uptake, Figure 4 shows immunostaining for Vax1 protein on axons and in D in the immune-EM figures, only on the axon surface; this is puzzling as the authors' model is that Vax1 is internalized and thus one would expect Vax1 to be in the cytoplasm of the cell *–* axon or cell body.

8) In Figure 9—figure supplement 1, the authors show that Vax1 and Slit are mutually antagonistic with respect to axon outgrowth. In the images (S9C), it appears that the addition of Slit blocks Vax1 localization in the axons to some extent. This should be examined in biochemical assays by assessing how the presence of Slit or Vax1 affects the other's binding to HSPGs (similar to assays in Figure 7—figure supplement 1). Alternatively, it would be informative to know whether addition of Slit inhibits Vax1 induction of axonal translation (Figure 8).

9) It is not clear whether soluble vax1 primarily exerts its function as a growth promoting factor (chemotrophic, i.e. Figure 5) or steering function (chemotropic, i.e. Figure 1) or both? Can the effects of vax1 secreted from COS7 cells be blocked in the presence of the anti-vax1 serum?

10) Blocking of translation with anisomycin may affect neurite outgrowth in a vax1-independent manner. Additional controls are needed to show that in the presence of a growth promoting cue (other than vax1) blocking of translation with anisomycin does not influence outgrowth rate.

[Editors’ note: the reviewers had the following concerns after re-review, which were addressed before acceptance.]

We are very sorry to report that, after a second round of review, the changes you made were not deemed to sufficient to render your paper acceptable for publication in *eLife* at this point. We hope that the detailed comments of the reviewers will help you in improving the paper.

Reviewer 1:

The authors have addressed the reviewers' comments, performing additional experiments and presenting new micrographs and other data. Nonetheless, there are still some amendments needed to clarify the model and data.

1) The reader must still work very hard in understanding how Vax acts to enhance axon growth.

a) One problem is that the authors “tell the story” in different ways, sometimes completely and in other places, leaving out one of the elements.

In the rebuttal: “Our overall hypothetical model is that HSPGs capture secreted Vax1 and release it to the plasma membrane for further cytoplasmic penetration.”

“This unconventional axon growth factor also utilizes HSPGs for anchoring to RGC axons and antagonizes Slit effects on RGC axon binding and growth (Figure 9—figure supplement 1).”

In the rebuttal, second paragraph: “We modified our model (revised Figure 9) regarding Vax1 and Slit competition for HSPGs in RGC ... in this competition, vHT-enriched Vax1 displaces Slit, which is bound to HSPGs in RGC axons at the lateral diencephalon, and promotes further growth toward the midline where RGC axons meet pathway determinants (e.g., VEGF164”, etc).

b) The all-important schema graphically illustrating the steps, in Figure 9, does not clearly illustrate what the authors intend to describe. From the graphic, one cannot intuit the process of Vax attachment to HSPGs on RGC surface, then enhancement of growth during midline crossing.

The various interactions might be broken into steps (attachment of Vax to HSPGs, then internalization, followed by retrograde transport to the cell body and/or action locally in the growth cone, finally axon extension)?

The growth cone should be enlarged, and perhaps use a different icon for the guidance molecules.

Sema6D is not a repellant for ipsilateral axons; it belongs with NrCAM on the contralateral axons.

2) In the rebuttal, major point 7: The co-culture assays with knockout vHT and Robo1-Fc should be included in the manuscript; these are important data and should not be hidden only for reviewers' inspection.

3) Response to minor point 7 is a little dismissive and unsatisfactory. The DiI labeling in the images of cryosectioned DiI is so fuzzy/overexposed/difficult to see clearly, that they would be better off just using the cytoarchitecture to determine the location of the RGC axon bundle. As is, the DiI quality distracts from the overall point of the figure. If the immunostaining is more important for them to show, as they indicate, they should focus on that and orient things based on cytoarchitecture, which should be sufficient to locate the axon bundles, and just leave out the DiI altogether.

4) In minor point 10 re: Figure 2: the authors don't address concerns about the quality of the Myc-GFP staining or show higher mag images.

5) In minor point 10 about Figure 2: is it not acceptable to run a negative control on a separate gel.

6) Response to minor point 10, Figure 3. Increasing the concentration of until one sees a response is a bit iffy. The explanation is confusing.

7) Likewise, the response to minor point 10, Figure 7 is unsatisfactory. The difficulties in getting good bands with some proteins and antibodies in Western blots is acknowledged. However, these are really very difficult to see and if the authors ran it 4 times and these are the best looking bands, the accuracy and reliability of these bands are questionable. Readers should not have to squint to tell whether or not there is a band present. The images are very grainy and pixelated.

8) The response to Major point 9 is “hand-wavy”... Are the authors trying to say that, yes, they think Vax1 actions are both chemotrophic and chemotropic, or are they undecided? As it currently reads, these statements are weak.

Reviewer 2:

Overall, I think the manuscript is much improved, particularly with respect to grammar and inclusion of appropriate references and added discussion points. The authors have addressed most of the concerns to an acceptable level, and I would support the acceptance of the paper, pending addressing the following points:

Regarding substantive concerns point 2: Although NrCAM and VegfA are expressed in the Vax1-/- chiasm, the morphology of the chiasm is clearly abnormal (Figure A.B and Supplemental Figure 1). In the text, the authors state: “The development of these OC-forming cells was normal in Vax1-deficient (Vax1-/-) mice (2) (Figure 9 1A,B; bottom rows), suggesting that the impairment in OC development in Vax1-deficient mice is not caused by the lack of OC-forming vHT cells but rather by a loss of RGC attractant molecules in these cells ”. It would be more accurate to state: “Although the morphology of the chiasm is abnormal in Vax1-/- deficient mice, the OC-forming cells and several chiasm localized cues (NrCAM, VegfA) are present.”

Also regarding substantive concerns point 2, with respect to the point about HSPGs functioning as co-receptors for other secreted proteins, I think the authors missed the reviewers' point. If I understand correctly, they were asked to consider whether Vax1/HSPGs co-localize with known cell surface receptors that may mediate internalization of Vax1. In addition, the co-localization of Vax1 and Sdc3 in RGC axons in tissue sections is pretty dirty and the co-localization isn't very convincing. Have the authors tried this in cultured RGCs? This might improve the quality of the images.

Regarding substantive concerns point 5: Despite the fact that Vax2 is not secreted, the ability of recombinant Vax2 to be internalized and induce axonal growth strengthens the findings of the paper. These data should be included in the manuscript as a supplemental figure, at the least.

Regarding substantive concerns point 8: While the authors nicely showed that Vax1 and Slit compete with one another for binding to HSPGs, they did not show whether addition of Slit blocks Vax1's ability to induce translation in axons. Did the authors do this experiment?

Regarding substantive concerns point 9: The authors should include the data showing that the Vax1 antibody blocks the function of Vax1 secreted from Cos7 cells as supplemental data.

Regarding minor concern 7: If this is the best image for IP experiments, perhaps the authors could complement the experiments by using overexpression of tagged constructs for Sdc2, Sdc3, Pax2 and Glp1 or by performing the reverse IP if antibodies are available.

In general, the proposed model of Vax1 action in RGC axons is still a bit confusing. Does it primarily function as a promoter of axonal growth via increased translation, or by competing with Slit binding to HSPGs? While these need not be mutually exclusive, I'm still a little unclear as to what the Slit/Robo data adds to the model, since Slit doesn't function as a general inhibitor of growth, but rather functions as a repulsive cue from the surrounding tissue.

---

## [Author Response]

*1) The references to other studies are deficient (missing or improper citation/interpretation) and the text should be edited for proper English usage. For example, the authors should cite papers regarding the idea that HSPGs are involved in axon guidance, and particularly at the optic chiasm, since this is not a new concept. For example:*
[8]*, J Comp Neurol 436: 236;*
[35]*, J Neurosci.26:6911; Conway et al., 2011, J Neurosci 31: 1955; Wilson and Stoeckli, 2013, Neuron 79:478; Sánchez-Arrones et al., 2013, J Neurosci 33:8596; Fabre et al., 2010, J Neurosci 30:266 and Yam and Charron, Curr Op Neurobiol 2013)*.

The manuscript was streamlined, reflecting revisions, and edited by multiple native-English speakers, including a professional scientific editor. As suggested by the reviewers, we have also added references relating to the functions of HSPGs as co-receptors for secreted.

*2) Although the authors show that the chiasm appears normal by RC2 staining, they should confirm that the expression of known chiasm localized cues (Vegf, ephrinB1, NrCAM) is normal*.

We examined expression of two vHT RGC attractive cues, *Vegfa* and NrCAM, by *in situ* hybridization and immunostaining, respectively. The results indicated that these chiasm cues are still expressed in E14.5 *Vax1-ko* mouse embryonic ventral diencephalon (Figure 1—figure supplement 1). These are even expanded more dorsally, suggesting that defective midline access of *Vax1-ko* mouse RGC axons was not resulted from the loss of these attractive chiasm cues.

*Similarly, there is no information on localization of HSPGs on RGCs nor is there any proposal/analysis on the fate of HSPGs after Vax1 binds: are they also internalized? If not, are they cleaved once bound to Vax1? These points should be discussed*.

We investigated the distribution of Vax1 and Sdc3 in E14.5 mouse retinas by co-immunostaining. As shown in Figure 7 and Figure 7—figure supplement 2, Vax1 and Sdc3 are co-localized in RGC axons, whereas these co-localized Vax1 signals disappeared after treating retinal sections with heparinase I. These results suggest that Vax1 interacts with Sdc3 in a HS-dependent manner at the cell surface, but is unlikely internalized by Sdc3-mediated endocytosis, which can protect Vax1-HS interaction from heparinase. We were also not able to detect any cleavage of Sdc2 or Vax1 in transfected cells. We provide the results for the reviewers’ inspection only. Our overall hypothetical model is that HSPGs capture secreted Vax1 and release it to the plasma membrane for further cytoplasmic penetration.

*A large number of HSPG binding proteins have been identified (e.g. FGF2, slit2, Semas, ephrins, syndecans, glypicans, etc). For all these cues HSPGs function as co-receptors for a transmembrane spanning signaling receptor. The authors should discuss this possibility and whether there is data for co-localization of these on RGCs*.

We discussed possible roles of HSPGs as co-receptors for netrin, Shh, and Slit during axonal projection of spinal commissural neurons and RGCs (please see the first paragraph of Discussion). We did not check the co-localization of these cues on RGCs. We have also not found any studies reporting co-localization of these cues on RGC axons, although previous results suggest that HSPGs are required for the actions of netrin and Slit on RGC axons (18; 31).

*3) The explants tested were taken from dorsal retina, at E13, when a few cells project ipsilaterally but ultimately disappear. The ventrotemporal retina gives rise to the permanent ipsilateral component after E14. Have the authors analyzed expression of syndecan or glypican in the “ipsi” retina and how Vax1-HSPG interactions underlie ipsi axon midline avoidance? This could be addressed by comparing ventrotemporal versus dorsotemporal retina*.

We examined the distribution of Sdc3 in E14.5 retinas, and found no region-specific enrichment of Sdc3 in the nasal-temporal (N/T) axis (Sdc3 images in Figure 7—figure supplement 2). These results are consistent with a previous report showing a uniform distribution of HSPGs in RGCs along the N/T axis (8).

We also tested axonal responses of retinal quadrants to Vax1. We provide the results for the reviewers’ inspection. The results suggest that ventral retinal axons respond to Vax1 as efficiently as those from the dorsal retinal explants. These results also show that V/T retinal axons express Sdc3 and project as efficiently as those from the dorsal retinal explants upon incubating with Vax1.

Therefore, as we propose in our model (Figure 9), Vax1 supports the projection of RGC axons toward the vHT midline regardless of their original positions in the retina. In the vHT midline, EphB1-expressing V/T RGC axons are repelled by ephrinB2-expressing radial glia while the remaining RGC axons continue growing in the contralateral direction.

4) The authors propose that vax1 overrides Robo/slit repulsion. Is slit repulsion of cultured RGC axons reduced (blocked) in the presence of recombinant vax1?

Yes. Slit and Vax1 competitively inhibit each other’s effects as well as their binding to RGC axons (please see revised Figure 9—figure supplement 1).

*Slits are thought to inhibit RGC axons from straying (defasciculating) from the main bundles at the chiasm, in an “inhibition-surround” mechanism, not as an inhibitory molecule at the crossing point (see Erskine et al., J Neurosci 2000, 20:4975;*
[34]
*Neuron 33:219)*.

As the reviewers indicate, Slit-Robo signaling inhibits RGC axons from exiting the optic track and projecting toward ectopic sites, such as the POA and contralateral OS (13; 34). This suggests that the role of Slit-Robo signaling in RGC axons is related to reducing the dynamics of the RGC axon growth cone rather than specifically enhancing fasciculation, which is still normal in the OS of *Slit1/2-ko* mice (please see Figure 3 of [34], Neuron 33: 219).

*The authors placed Vax-1-containing cells in the third ventricle above the chiasm in an isolated prep, showing that growth was enhanced, and application of Fc-Robo3 also enhanced RGC axon growth (together growth was further enhanced). While this experiment argues for Slit involvement in this preparation, it cannot relate to guidance in vivo because Robo3 is not expressed in the mouse retina (see above references and Thompson et al., 2006, J Neurosci 26:8092). Further, there should be axon extension in the control preparation in*
Figure 9.

As the reviewers’ indicated, Robo3 is not expressed in mouse RGCs, where Robo1 and Robo2 are consistently expressed (13). In addition, Robo3-Fc alone was not effective in restoring RGC axon growth in *Vax1-ko* mouse embryos (original Figure 9).

In the revised manuscript, we therefore replaced the original results with those using Robo1-Fc. As shown in revised Figure 9, implanted Robo1-Fc induced RGC axon regrowth in *Vax1-ko* mouse brains and it further facilitated the effects of co-implanted Vax1.

*Finally, the model posed in*
Figure 9
*is misleading: how would the Vax1-HSPG interaction act on Slit inhibition leading to action of molecules important for contralateral crossing (e.g., VEGF164, NrCAM, and Semaphorin6D, latter, erroneously indicated for ipsilateral growth)?*

We modified our model (revised Figure 9) regarding Vax1 and Slit competition for HSPGs in RGC axons and antagonistic regulation of local translation. Accordingly, in this competition, vHT-enriched Vax1 displaces Slit, which is bound to HSPGs in RGC axons at the lateral diencephalon, and promotes further growth toward the midline where RGC axons meet pathway determinants (e.g., VEGF164, NrCAM, Semaphorin6D, and ephrinB2).

*5) The authors show that in contrast to Vax1, Vax2 is not secreted and does not bind to HSPGs (*Figure 2*,*
Figure 7—figure supplement 1*). This is somewhat surprising, since the domains that regulate secretion of homeodomain transcription factors are typically within the homeodomain region, which is highly conserved between Vax1 and Vax2. Have the authors attempted to identify the specific domains required for secretion and HSPG binding that differ between Vax1 and Vax2?*

According to our unpublished results, only about 40% of homeodomain transcription factors can be secreted at least in one cell type (*Lee et al., manuscript in preparation*). It suggests that secretion of homeodomain proteins is not solely dependent of homeodomain but rely on their three-dimensional structures. Therefore, in COS7 cells, Vax2 might fail to form a three-dimensional structure that can be recognized by secretion machineries that have not been identified yet.

We previously showed that Vax2 can be localized to the cytoplasm of the developing mouse retina (22). This cytoplasmic retention of Vax2 is mediated by phosphorylation at S170, which is absent in Vax1. However, replacing of this phosphorylation site by alanine (S170A) or aspartate (S170D) was failed to induce secretion of the Vax2 proteins (data not shown). It suggests that the phosphorylation-induced cytoplasmic retention of Vax2 is unlikely related with secretion.

*The authors should test whether the addition of exogenous Vax2 can bind to RGCs and promote their outgrowth using assays like those used in*
Figure 3.

Bacterial-expressed recombinant Vax2 was able to induce RGC axon growth after penetration. We provide the results for reviewers’ inspection only. This suggests that Vax2, which shares an identical cell-penetration sequence with Vax1, is functionally equivalent to Vax1 *in vitro* but is simply not secreted from COS7 cells.

6) Immuno-EM studies show that vax1 is localized to the axonal surface of RGCs and also found in endocytosed vesicles. To participate in local translation, vax1 likely needs to be in the cytoplasm not inside vesicles?

In the revised manuscript, we provide additional iTEM images showing cytoplasmic Vax1 (please see Figure 4—figure supplement 2).

*7) In the co-culture assays in*
Figure 1*, WT retinal explants cultured with Vax1-/- vHT not only shows no attraction, but appears to show active repulsion. This may be due to the secretion of Slit from the explants, which should be tested by examining whether addition of Robo-Fc (similar to*
Figure 9*) inhibits this repulsion*.

As recommended by the reviewers, we tested the effects of Robo1-Fc on RGC axon growth from retinal explants co-cultured with vHT explants. We provide the results for reviewers’ inspection. We found that Robo1-Fc treatment stimulated RGC axonal projection, not only toward *Vax1-ko* vHTs but also in other directions. Therefore, consistent with the reviewers’ hypothesis, Slit from the hypothalamic explants repels RGC axons more efficiently in the absence of Vax1.

*With respect to Vax localization and uptake,*
Figure 4
*shows immunostaining for Vax1 protein on axons and in D in the immune-EM figures, only on the axon surface; this is puzzling as the authors' model is that Vax1 is internalized and thus one would expect Vax1 to be in the cytoplasm of the cell – axon or cell body*.

Although a majority of Vax1 bound to the RGC membrane, a significant amount of Vax1 protein was also detectable in the RGC cytoplasm. This cytoplasmic Vax1 protein might be sufficient to exert axon growth-stimulatory effects. In the revised manuscript, we provide additional iTEM images showing cytoplasmic Vax1 (please see images in Figure 4—figure supplement 2).

*8) In*
Figure 9—figure supplement 1*, the authors show that Vax1 and Slit are mutually antagonistic with respect to axon outgrowth. In the images (S9C), it appears that the addition of Slit blocks Vax1 localization in the axons to some extent. This should be examined in biochemical assays by assessing how the presence of Slit or Vax1 affects the other's binding to HSPGs (similar to assays in*
Figure 7—figure supplement 1*). Alternatively, it would be informative to know whether addition of Slit inhibits Vax1 induction of axonal translation (*Figure 8*)*.

To address the competition for HSPG binding between Vax1 and Slit, we added Vax1 and Slit2 proteins under the same conditions tested in Figure 9—figure supplement 1. The results indicated that Vax1 and Slit2 compete for binding to RGC axons (Figure 9—figure supplement 1). These results therefore suggest that Vax1 and Slit proteins compete for binding to the sugar chain in the extracellular domain of HSPGs. However, a possible antagonism between Vax1 and Robo signaling in enhancing the translation of specific target mRNAs cannot be excluded at this point. Therefore, we propose a model in which Vax1 antagonizes Slit-Robo signaling not only at the level of ligand accumulation but also at the level of intracellular signaling (Figure 9).

*9) It is not clear whether soluble vax1 primarily exerts its function as a growth promoting factor (chemotrophic, i.e.*
Figure 5*) or steering function (chemotropic, i.e.*
Figure 1*) or both?*

Vax1 can also play roles as a steering factor in certain context, which Vax1 is expressed in a specific region. In mouse optic pathway, Vax1 is expressed in OS as well as vHT (2; 15). It makes RGC axons be consistently exposed to Vax1 until it reaches to optic tract. In addition, Vax1 not only works in RGC axon terminal, but it is also present in axon shaft to promote axon growth (Figure 8). Therefore, we propose Vax1 is an axon growth factor in general.

Can the effects of vax1 secreted from COS7 cells be blocked in the presence of the anti-vax1 serum?

We provide results showing that anti-Vax1 antibody sequestered secreted Vax1 from COS7 cells and blocked the transfer of Vax1 into retinal axons. We provide the results for reviewers’ inspection.

*10) Blocking of translation with anisomycin may affect neurite outgrowth in a vax1-independent manner. Additional controls are needed to show that in the presence of a growth promoting cue (other than vax1) blocking of translation with anisomycin does not influence outgrowth rate*.

As the reviewers indicated, anisomycin does not specifically affect Vax1; it also inhibits global translation, which is essential for RGC axon growth induced by other cues (Lin &Holt, 2007). Accordingly, since results obtained with anisomycin may be misleading, we eliminated them from the revised manuscript.

[Editors’ note: the reviewers had the following concerns after re-review, which were addressed before acceptance.]

*We are very sorry to report that, after a second round of review, the changes you made were not deemed to sufficient to render your paper acceptable for publication in eLife at this point. We hope that the detailed comments of the reviewers will help you in improving the paper*.

As you can find from the comments of our reviewers, they mostly indicate an unclear explanation about the function of secreted Vax1 between an axon growth promoting factor and a steering factor. I admit that we might not have understood the reviewers’ questions fully and answered insufficiently in our rebuttal. Now, we understand it exactly and can answer clearly that “Vax1 is a novel axon growth promoting factor, which works in RGC axoplasm to activate local protein synthesis, than a steering factor that mainly acts on specific receptors to trigger on intracellular signaling cascades.” This will also be further clarified in the text and by drawing a separate model diagram with details as reviewer #1 proposes.

Reviewer 1:*The authors have addressed the reviewers' comments, performing additional experiments and presenting new micrographs and other data. Nonetheless, there are still some amendments needed to clarify the model and data*.

*1) The reader must still work very hard in understanding how Vax acts to enhance axon growth*.

*a) One problem is that the authors “tell the story” in different ways, sometimes completely and in other places, leaving out one of the elements*.

In this paper, we propose Vax1 as a novel RGC axon growth factor that binds to HSPGs and enters into RGC axoplasm to trigger local translation. We believe this message is consistent throughout the paper, although it is expressed with different emphases in each paragraph. However, we modified or eliminated expressions that could make our readers difficult to understand the conclusion correctly.

*In the rebuttal: “Our overall hypothetical model is that HSPGs capture secreted Vax1 and release it to the plasma membrane for further cytoplasmic penetration*.*”*

This expression emphasizes roles of HSPGs that capture Vax1 for subsequent cell penetration but do not induce intracellular signaling cascades or endocytosis of Vax1. This is not a key part of our answer to the reviewer’s question that asks a potential endocytosis or cleavage of HSPG-interacting Vax1. We eliminated this sentence from the rebuttal.

*“This unconventional axon growth factor also utilizes HSPGs for anchoring to RGC axons and antagonizes Slit effects on RGC axon binding and growth (*Figure 9—figure supplement 1*)*.*”*

This sentence is a part of a paragraph that compares an orchestrated RGC axon growth to that of commissural axons of spinal cord. It describes the reciprocal antagonism between Vax1 and Slit in RGC axons as an analogous event of netrin (or Shh) and Slit in spinal commissural axons. In following this sentence, however, we emphasize the main activity of Vax1 in RGC axon as that ‘In addition, imported Vax1 promotes RGC axon growth by stimulating local protein synthesis (Figure 8)’. We rewrote the paragraph to avoid the misinterpretation that Vax1 mainly interferes with Slit-Robo interaction to support RGC axon growth.

*In the rebuttal, second paragraph: “We modified our model (revised*
Figure 9*) regarding Vax1 and Slit competition for HSPGs in RGC ... in this competition, vHT-enriched Vax1 displaces Slit, which is bound to HSPGs in RGC axons at the lateral diencephalon, and promotes further growth toward the midline where RGC axons meet pathway determinants (e.g., VEGF164”, etc)*.

This is the part of our answer to the reviewer’s question related to the effects of Vax1 on contralateral pathway determination cues (e.g., VEGF164, NrCAM, and Semaphorin6D). We added to this sentence to emphasize that Vax1 does not influence to the expression of these pathway determination cues (please see Figure 1—figure supplement 1) but promotes RGC axons to grow to the midline where these cues are expressed. However, the expression like ‘*vHT-enriched Vax1 displaces Slit…*’ might lead to misunderstanding that main activity of Vax1 is competing with Slit. Thus, we rewrote our rebuttal by eliminating the part expressed improperly.

*b) The all-important schema graphically illustrating the steps, in*
Figure 9*, does not clearly illustrate what the authors intend to describe. From the graphic, one cannot intuit the process of Vax attachment to HSPGs on RGC surface, then enhancement of growth during midline crossing*.

We presented this simplified model owing to limited space as a sub-figure, i.e. Figure 9. We present a detailed diagram in a separate figure, i.e. Figure 10, reflecting the reviewer’s requests in following.

The various interactions might be broken into steps (attachment of Vax to HSPGs, then internalization, followed by retrograde transport to the cell body and/or action locally in the growth cone, finally axon extension)?

*The growth cone should be enlarged, and perhaps use a different icon for the guidance molecules*.

*Sema6D is not a repellant for ipsilateral axons; it belongs with NrCAM on the contralateral axons*.

We apologize for omitting Sema6D in the contralateral cues. Sema6D is indeed a context-dependent cue for RGC axons rather than simple an attractive or repulsive cue as Erskine *et al.* (2013) showed. It repels RGC axons in the absence of NrCAM and Plexin A1. To avoid confusion, we simplified the attractive cues as NrCAM and VEGF164 in our revised model diagram (Figure 10).

*2) In the rebuttal, major point 7: The co-culture assays with knockout vHT and Robo1-Fc should be included in the manuscript; these are important data and should not be hidden only for reviewers' inspection*.

Following the reviewer’s suggestion, we included this figure in the paper. The reason that we provided this for reviewers’ inspection only was not for hiding but for making the paper compact. The original paper already had a large body with 9 figures, 9 figure supplements, and 2 video files. We were afraid that our readers would be overwhelmed by the large amount of data. In the revised paper, we provide all “reviewers’ inspection only” figures as figure supplements.

*3) Response to minor point 7 is a little dismissive and unsatisfactory. The DiI labeling in the images of cryosectioned DiI is so fuzzy/overexposed/difficult to see clearly, that they would be better off just using the cytoarchitecture to determine the location of the RGC axon bundle. As is, the DiI quality distracts from the overall point of the figure. If the immunostaining is more important for them to show, as they indicate, they should focus on that and orient things based on cytoarchitecture, which should be sufficient to locate the axon bundles, and just leave out the DiI altogether*.

We removed DiI fluorescence images from dark field images in the figures as the reviewer suggests. We also redraw a schematic diagram of slab experiment in Figure 4 (please see a drawing in top row).

*4) In minor point 10 re:*
Figure 2*: the authors don't address concerns about the quality of the Myc-GFP staining or show higher mag images*.

We did not use Myc-GFP for control but use 3X-Myc tag in this figure. We have already provided a negative control for anti-Myc antibody on top row (Mock).

*5) In minor point 10 about*
Figure 2*: is it not acceptable to run a negative control on a separate gel*.

We replaced these with new Western blot bands shown in a same gel.

*6) Response to minor point 10,*
Figure 3*. Increasing the concentration of until one sees a response is a bit iffy. The explanation is confusing*.

We explained more clearly in the revised rebuttal. We also provide a video recorded retinal axons treated with 6X-His-FITC peptide at same weight of Vax1-His-FITC protein (please see Video 3).

*7) Likewise, the response to minor point 10,*
Figure 7
*is unsatisfactory. The difficulties in getting good bands with some proteins and antibodies in Western blots is acknowledged. However, these are really very difficult to see and if the authors ran it 4 times and these are the best looking bands, the accuracy and reliability of these bands are questionable. Readers should not have to squint to tell whether or not there is a band present. The images are very grainy and pixelated*.

We repeated the experiments with rabbit anti-Glp1 and anti-Sdc3 antibodies from different company sources, and replaced those with new results with better qualities (please see revised Figure 7).

*8) The response to Major point 9 is “hand-wavy”... Are the authors trying to say that, yes, they think Vax1 actions are both chemotrophic and chemotropic, or are they undecided? As it currently reads, these statements are weak*.

Although we believed Vax1 is mainly a chemotrophic factor, which stimulates RGC axon growth in a non-direction selective manner, we could not rule out a possible chemotropic activity in a certain condition such that Vax1 can bind to unidentified receptors that turn on intracellular signal cascades. However, our current results (Figure 8) decline the latter chemotropic activity of Vax1. We therefore answered more clearly by changing our rebuttal (please see our revised rebuttal to Substantive concern #9).

Reviewer 2: *Overall, I think the manuscript is much improved, particularly with respect to grammar and inclusion of appropriate references and added discussion points. The authors have addressed most of the concerns to an acceptable level, and I would support the acceptance of the paper, pending addressing the following points:*

*Regarding substantive concerns point 2: Although NrCAM and VegfA are expressed in the Vax1-/- chiasm, the morphology of the chiasm is clearly abnormal (Figure A.B and Supplemental Figure 1). In the text, the authors state: “The development of these OC-forming cells was normal in Vax1-deficient (Vax1-/-) mice (*[2]*) (Figure 9 1A,B; bottom rows), suggesting that the impairment in OC development in Vax1-deficient mice is not caused by the lack of OC-forming vHT cells but rather by a loss of RGC attractant molecules in these cells ”. It would be more accurate to state: “Although the morphology of the chiasm is abnormal in Vax1-/- deficient mice, the OC-forming cells and several chiasm localized cues (NrCAM, VegfA) are present*.*”*

We appreciate the reviewer’s suggestion and changed the expression in the text.

*Also regarding substantive concerns point 2, with respect to the point about HSPGs functioning as co-receptors for other secreted proteins, I think the authors missed the reviewers' point. If I understand correctly, they were asked to consider whether Vax1/HSPGs co-localize with known cell surface receptors that may mediate internalization of Vax1. In addition, the co-localization of Vax1 and Sdc3 in RGC axons in tissue sections is pretty dirty and the co-localization isn't very convincing. Have the authors tried this in cultured RGCs? This might improve the quality of the images*.

We replaced these with co-staining images with Vax1/NF160/Sdc3 in Figure 7.

*Regarding substantive concerns point 5: Despite the fact that Vax2 is not secreted, the ability of recombinant Vax2 to be internalized and induce axonal growth strengthens the findings of the paper. These data should be included in the manuscript as a supplemental figure, at the least*.

We included the results as Figure 3—figure supplement 3.

Regarding substantive concerns point 8: While the authors nicely showed that Vax1 and Slit compete with one another for binding to HSPGs, they did not show whether addition of Slit blocks Vax1's ability to induce translation in axons. Did the authors do this experiment?

We have done the experiments but failed to observe the inhibition of translation (data not shown). It might be related with Slit-Robo also induces local translational for RGC axon growth cone repulsion.

*Regarding substantive concerns point 9: The authors should include the data showing that the Vax1 antibody blocks the function of Vax1 secreted from Cos7 cells as supplemental data*.

We included the results as Figure 3—figure supplement 2.

*Regarding minor concern 7: If this is the best image for IP experiments, perhaps the authors could complement the experiments by using overexpression of tagged constructs for Sdc2, Sdc3, Pax2 and Glp1 or by performing the reverse IP if antibodies are available*.

We replaced these with new results with better qualities (please see revised Figure 7).

*In general, the proposed model of Vax1 action in RGC axons is still a bit confusing. Does it primarily function as a promoter of axonal growth via increased translation, or by competing with Slit binding to HSPGs? While these need not be mutually exclusive, I'm still a little unclear as to what the Slit/Robo data adds to the model, since Slit doesn't function as a general inhibitor of growth, but rather functions as a repulsive cue from the surrounding tissue*.

We apologize to leave this part still unclear. This can be clarified by removing unnecessary sentences that we had added in the rebuttal as well as in the manuscript. As we answered to reviewer #1, the main activity of Vax1 for RGC axon growth is a promoter of axonal growth via local protein synthesis than HSPG binding. We also redrew a hypothetical model diagram by explaining HSPGs binding of Vax1, cell penetration, and stimulating local translation in detail (please see Figure 10).